# Precise mycobacterial species and subspecies identification using the PEP-TORCH peptidome algorithm

Duran Bao [1,2,4], Sudipa Maity[1,2,4], Lingpeng Zhan [1,2,4], Seungyeon Seo [3], Qingbo Shu[1,2], Christopher J Lyon[1,2], Bo Ning[1,2], Adrian Zelazny[3], Tony Y Hu [1,2] & Jia Fan [1,2]✉

## Abstract

Mycobacterial infections pose a significant global health concern, requiring precise identification for effective treatment. However, diagnosing them is challenging due to inaccurate identifications and prolonged times. In this study, we aimed to develop a novel peptidome-based method using mycobacterial growth indicator tube (MGIT) cultures for faster and more accurate identification. We created the PEPtide Taxonomy/ORganism CHecking (PEP-TORCH), an algorithm that analyzes tryptic peptides identified by mass spectrometry to diagnose species and subspecies with predominance scores. PEP-TORCH demonstrated 100% accuracy in identifying mycobacterial species, subspecies, and co-infections in 81 individuals suspected of mycobacterial infections, eliminating the need for a sub-solid culture procedure, the gold standard in clinical practice. A notable strength of PEP-TORCH is its ability to provide information on species and subspecies simultaneously, a process conventionally achieved sequentially. This capability significantly expedites pathogen identification. Furthermore, a targeted proteomics method was validated in 63 clinical samples using the taxa-specific peptides selected by PEP-TORCH, making them suitable as biomarkers in more clinically friendly settings. This comprehensive identification approach holds promise for streamlining treatment strategies in clinical practice.

**Keywords** Algorithm; LC-MS/MS; Nontuberculosis Mycobacterium; Peptidome; Subspecies
**Subject Categories** Computational Biology; Methods & Resources; Microbiology, Virology & Host Pathogen Interaction

## Introduction

Mycobacterial infections, encompassing a broad spectrum of nontuberculous mycobacteria (NTM) alongside the *Mycobacterium tuberculosis* complex, pose an escalating global public health challenge (Bhanushali et al, 2023; Zhang et al, 2022). Global studies consistently highlight the presence of geographic heterogeneity in NTM distribution and disease patterns across different regions. In North America, both national and subnational studies consistently reveal a rising prevalence and incidence of NTM isolation and disease (Prevots et al, 2023). Florida is one of the NTM hotspots of the United States, showing the disease prevalence increased from 14.3/100,000 in 2012 to 22.6/100,000 in 2018 (Kambali et al, 2021). The NTM are opportunistic pathogens widely distributed throughout environmental habitats and have varied geographical distributions (Donohue, 2021). According to the American Lung Association, the majority of NTM lung infections are caused by *Mycobacterium avium*, *Mycobacterium intracellulare*, *Mycobacterium kansasii*, and *Mycobacterium abscessus*. In both tuberculosis (TB) and NTM pulmonary diseases, the bacterial characteristics and the host factors influence the susceptibility and manifestations of infection as well as the outcome of treatment. NTM produces symptoms similar to TB but requires distinct drug regimens. Thus, accurate NTM species or subspecies identifications are crucial for successful management (Shu et al, 2022a).

The gold standard protocol for mycobacterial detection typically begins with a liquid culture media in mycobacterial growth indicator tube (MGIT). The culture flagged positive for mycobacteria by the MGIT instrument is further sub-cultured in solid media for a longer period before using standard approaches like matrix-assisted laser desorption ionization time-of-flight mass spectrometry (MALDI-TOF MS) on the culture (Rotcheewaphan et al, 2019). MALDI-TOF MS provides valuable preliminary information to guide treatment options, but often cannot distinguish closely related species, has limited ability to detect drug resistance, and does not detect and identify specific proteins (Hou et al, 2019; Rodriguez-Sanchez et al, 2016). In addition, MALDI-TOF MS on a liquid medium yields poorer results than those using solid medium (Rodriguez-Temporal et al, 2018). Polymerase chain reaction (PCR) for mycobacterial identification is on the rise; however, it preferably requires a solid culture medium and cannot identify co-infections (Uwamino et al, 2023).

[1]Center for Cellular and Molecular Diagnostics, Tulane University School of Medicine, New Orleans, LA 70112, USA. [2]Department of Biochemistry and Molecular Biology, Tulane University School of Medicine, New Orleans, LA 70112, USA. [3]Department of Laboratory Medicine, NIH Clinical Center, NIH, Bethesda, MD 20892, USA. [4]These authors contributed equally: Duran Bao, Sudipa Maity, Lingpeng Zhan. ✉E-mail: jfan5@tulane.edu

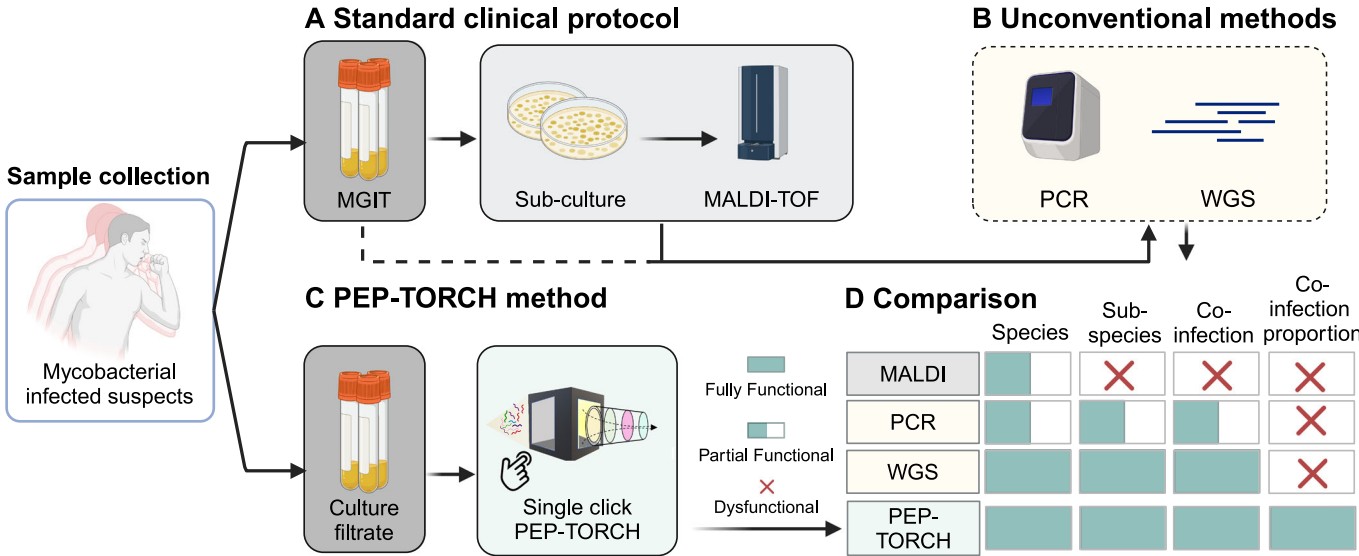

**Figure 1. Comparison of workflows for the identification of mycobacteria.**

(A) Samples collected from mycobacterial infected suspects were cultured in mycobacterial growth indicator tubes (MGIT), and on being indicated positive by the BACTEC MGIT 960 instrument, bacterial colonies from liquid medium were further sub-cultured before being identified by the current clinical diagnostic standard method, such as MALDI-TOF MS. (B) Other non-conventional methods that are infrequently used include PCR and WGS. (C). Our workflow utilizes culture filtrate from direct MGIT samples to analyze peptides and provide mycobacterial identification. (D) Compared to current standard methods, our method outperformed current clinical approaches by providing species, subspecies, co-infection status, and corresponding proportion at the same time without the need for further sub-culturing.

The precise identification of closely related pathogens via peptide sequence variations holds significant potential to improve diagnostic accuracy. Liquid chromatography tandem mass spectrometry (LC-MS/MS) has the sensitivity and resolution to detect specific sequence variations in peptides that can distinguish closely related taxa within the mycobacterium superfamily to permit specific diagnoses. However, translation of LC-MS/MS diagnostic applications into clinical laboratories has been hampered by the complexity of their sample preparation and data analysis methods. To overcome the limitations of detection of mycobacteria through prolonged solid culture methods, we developed a streamlined process to analyze culture filtrate protein (CFP) samples from MGIT growth cultures using LC-MS/MS, coupled with an automated PEPtide Taxonomy/ORganism CHecking (PEP-TORCH) pipeline. This pipeline identifies species- and subspecies-specific mycobacterial peptide signatures. A comparison of workflow between standard clinical protocols and the PEP-TORCH method is shown in Fig. 1. The MALDI-TOF Biotyper, the current clinical standard provides species- or species-complex-level identification; however, subspecies-level identification typically requires whole-genome sequencing (WGS). In contrast, PEP-TORCH delivers taxonomy scores for identified species and subspecies with reduced turnaround times by eliminating the need for solid culture. In this study, 102 samples (Table 1) were analyzed, with 81 validated by the PEP-TORCH pipeline. In addition, 63 samples underwent targeted proteomics guided by PEP-TORCH selection, including 42 samples validated by both methods and 21 exclusively through the targeted approach. The PEP-TORCH pipeline offers a comprehensive and efficient one-stop solution for species- and subspecies-level identification in NTM detection.

**Table 1. Demographic information of the clinical samples.**

| Sample source | Male | | Female | | | Total | |
|---|---|---|---|---|---|---|---|
| | n | Age mean (min–max) | n | Age mean (min–max) | n | | Age mean (min–max) |
| Sputum | 29 | 52 (17–87) | 39 | 54 (16–79) | 68 | | 54 (16–87) |
| Abscess | 1 | 44 | 2 | 39 (35–44) | 3 | | 41 (35–44) |
| Biopsy lung | 1 | 78 | | | 1 | | 78 |
| Biopsy lymph node | 1 | 27 | 2 | 50 (41–60) | 3 | | 43 (27–60) |
| Biopsy skin | 6 | 48 (23–64) | | | 6 | | 48 (23–64) |
| Bronchial wash | 2 | 48 (46–49) | 3 | 65 (46–80) | 5 | | 58 (46–80) |
| CSF | | | 1 | 61 | 1 | | 61 |
| Right eye anterior chamber | 1 | 78 | | | 1 | | 78 |
| Sinus | 1 | 53 | | | 1 | | 53 |
| Stool | | | 1 | 10 | 1 | | 10 |
| Bone marrow | 1 | 40 | | | 1 | | 40 |
| Wound | 3 | 23 (23–23) | | | 3 | | 23 (23–23) |
| Other Tissues | 2 | 27 (23–40) | | | 2 | | 27 (23–40) |
| BCG Vaccinated* | 1 | 27 | 6 | 31 (10–46) | 7 | | 30 (10–46) |
| Total | 48 | 49 (17–87) | 48 | 54 (16–80) | 96 | | 51 (16–87) |

*Detailed information on BCG vaccination is available in Dataset EV1A. BCG vaccination records are only applicable to *M. tuberculosis* cases. The sample numbers listed under BCG Vaccinated were not included in the total count, as they were already accounted for under their respective sample sources.

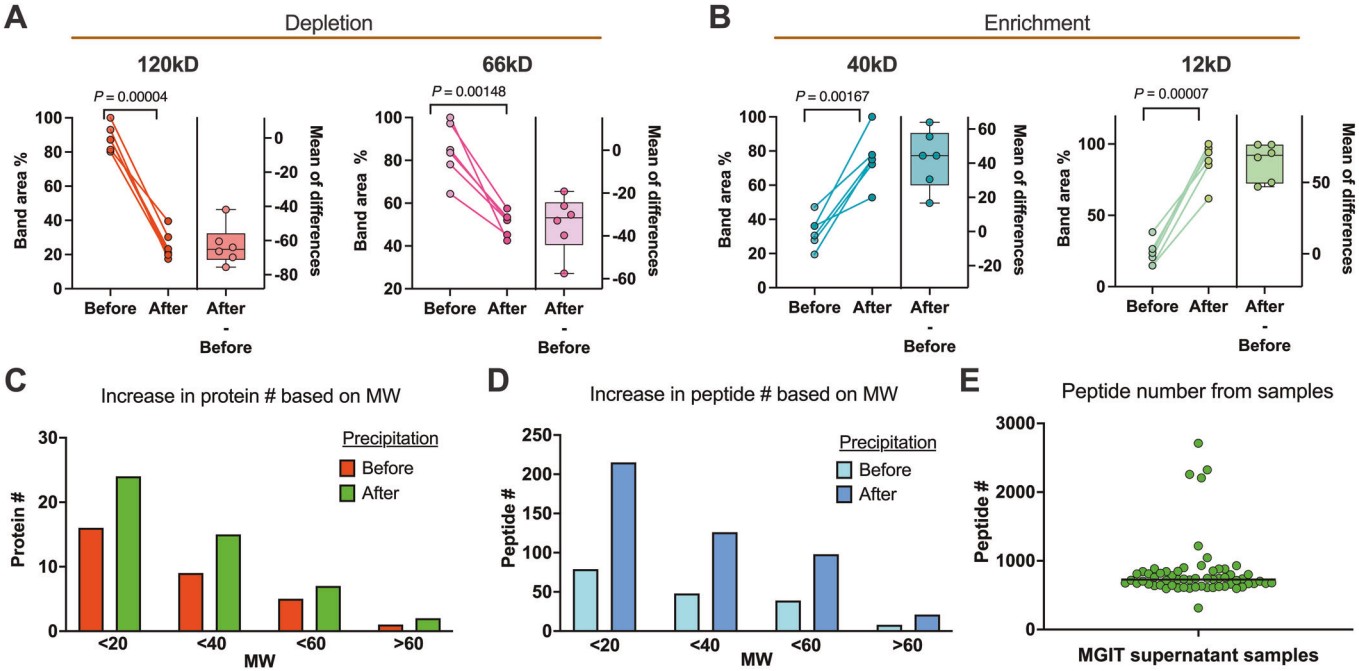

**Figure 2. Sample processing optimization.**

(A) Relative depletion of 120 and 66 kDa protein bands from media. (B) Relative enrichment of 12 and 40 kDa mycobacteria-derived protein bands after acetonitrile precipitation of CFP samples (*n* = 6), depicting individual band areas converted to a percentage scale with the largest band area set to 100. The corresponding box and whisker plot represents the differences in the bandwidths for each sample, with the box indicating the interquartile range and the whiskers representing the highest, mean, and lowest bandwidth differences. (C) The mean number of proteins mapped to the indicated size ranges. (D) Identified peptides within these size ranges as determined using MS data generated before and after acetonitrile precipitation for these six samples. Both protein and peptide number changes were significant with *P* = 0.002 and 0.0005, respectively. (E) In the scatter plot, the number of peptides identified in these samples (*n* = 62) belonging to each indicated species or complex, with their median represented by the black line in the middle. A two-tailed paired Student's *t* test was used for statistical analysis. Source data are available online for this figure.

# Results

## Acetonitrile precipitation of CFP samples improves mycobacteria proteome coverage

Acetonitrile precipitation significantly enhances mycobacterial proteome coverage in CFP samples, which were collected from MGIT culture flagged positive for mycobacterial growth by the BACTEC MGIT 960 instrument. The CFP samples were then subjected to an acetonitrile precipitation procedure (Kay et al, 2008) to deplete abundant high molecular weight (MW) proteins, enhancing LC-MS/MS detection of less abundant and lower MW mycobacteria-derived proteins. A significant reduction of 120 and 66 kDa proteins was observed following the precipitation step, possibly corresponding to major protein components of the MGIT media (Figs. 2A and EV1A), and a marked increase in two prominent 40 kDa and 12 kDa protein bands (Fig. 2B). Upon LC-MS/MS analysis, it was found that peptides retained in the supernatant after acetonitrile precipitation of these six sample pairs preferentially mapped to mycobacterial proteins <60 kDa and that these samples identified more proteins and peptides (64% ± 26 and 162% ± 8, respectively) (Fig. 2C,D and EV1B) as compared to samples before precipitation. Bottom-up data-dependent acquisition (DDA) proteomics and database searches (*Mycobacteriaceae*, taxonomy ID: 1762) identified a similar number of peptides

(729 ± 125) per sample (Fig. 2E; Dataset EV1A). This acetonitrile-fractionation method substantially improves the detection and analysis of mycobacterial proteins in CFP samples, demonstrating its efficacy in enhancing proteome coverage for a more comprehensive analysis of mycobacterial proteins in CFP samples.

## The PEP-TORCH algorithm classifies mycobacteria species and subspecies

LC-MS/MS data can detect minor sequence variations in peptides from homologous proteins (Fig. EV2) (Shu et al, 2022b). This feature is useful for distinguishing closely related mycobacterial isolates. However, employing this approach requires expertise in both MS and bioinformatics data analyses. To address these issues, we developed the automated PEP-TORCH algorithm, which employs a decision tree pipeline to screen tryptic peptides that can identify species and subspecies through a specific peptide-weighted (PW) scoring system (Fig. 3A).

The first step in the PEP-TORCH algorithm utilizes the Unipept application programming interface (API) code to perform a batch analysis of all peptides identified from the CFP samples within the R Studio platform. This step generates a peptide-species matching matrix, listing peptides and their corresponding matched organisms. The next step filters the matrix to exclude peptides matching organisms that either lack annotations related to mycobacteria or

## A Diagnostic output of samples analyzed by PEP-TORCH

## B Candidate filtering

## C Species and subspecies identification

## D Reproducibility

## E Co-infection

**Figure 3. Decision tree and algorithm of the pipeline.**

(A) Tryptic peptides identified by MS from MGIT filtrate samples were processed by PEP-TORCH to generate species and subspecies detection results, including peptide weightage scores. (B) PEP-TORCH processing includes filtering steps to exclude peptides that do not match mycobacteria or lack recorded human hosts in the Bacterial and Viral Bioinformatics Resource Center (BV-BRC). (C) The decision tree algorithm for taxon identification relies on single species-specific peptides, as well as combinations of two or three species-specific peptides, to determine species or subspecies. (D) Reproducibility testing on three replicates each from *M. tuberculosis* and *M. abscessus* achieved a perfect $PW_{sp}$ score of 100 for both species across all replicates when analyzing all identified peptides. Peptides identified across different replicates revealed that ~79% and 82% of peptides were common to all replicates of *M. tuberculosis* and *M. abscessus*, respectively. (E) Simulated mixed culture samples were prepared by mixing peptides in certain amount ratios (10:90, 25:75, 50:50, 75:25, and 90:10). These simulated subsets when analyzed in triplicates identified mixed species in these samples, and $PW_{sp}$ scores of the mixed samples were highly correlated with the peptide amount ratios. Pearson's correlation coefficients (r) between the scores and amount ratios ranged from 0.98 to 0.99. Source data are available online for this figure.

are not isolated from human hosts. The Bacterial and Viral Bioinformatics Resource Center database was used to verify the human host origin of the matched mycobacteria (Fig. 3B). The remaining matches are categorized based on their ability to identify species. As shown in Fig. 3C and examples in Appendix Table S1, PEP-TORCH identifies species-events through: (1) single specific peptides, which match only one species; (2) 2-peptide combinations, where overlapping matches from two peptides identify a single species; and

(3) 3-peptide combinations, where overlapping matches from three peptides identify a single species. PEP-TORCH limits multi-peptide combinations to three peptides, as further increasing the number of combinations does not significantly improve species or subspecies identification and adds a computational burden. For example, two cases illustrate this ability of the pipeline, showcasing the analysis of a single infection (ID: MAB4) and a co-infection sample (ID: Co-inf) where the infections were detected based on categorizing the unique

peptides or unique pairs of peptides. Searches performed to identify multi-peptide biomarker signatures were limited to three-peptide combinations as the increased computational workload required for these analyses did not yield significantly improved species or subspecies identifications. All species-specific peptides and peptide combinations for a sample were then compiled and used to calculate a PW species identification ($PW_{sp}$) score, by dividing the number of single and multi-peptide combinations that identify a specific mycobacterium by the total number of peptide combinations generated for that sample. Such $PW_{sp}$ scores were primarily determined by multi-peptide combinations derived from identified peptides (Dataset EV1B–F) since small numbers of peptides could form numerous permutations to generate extensive arrays of multi-peptide combinations (Dataset EV2). Sporadic detection of single peptide matches to discordant mycobacteria was therefore unlikely to decrease consensus $PW_{sp}$ scores, reduce diagnostic confidence, and alter the number of correct and incorrect identifications (Dataset EV2A–E). Simultaneously, secondary analyses were used to identify subspecies-specific single- or multi-peptide biomarkers (Dataset EV2F,G) and to calculate PW subspecies ($PW_{subsp}$) scores were determined in the same manner. For instance, for the single infection sample (Fig. 3A), MAB4 was identified as *M. abscessus* by 524 peptide combinations (Dataset EV3A) with a $PW_{sp}$ score of 100 at the species level, and 386 peptide combinations with a $PW_{subsp}$ score of 100 at the subspecies level. Such peptide combinations to determine species and subspecies identification were illustrated in Dataset EV3B–K.

Moreover, we investigated the reproducibility of both our MS and pipeline approach. To this end, we conducted experiments using three replicate samples from each species: *M. tuberculosis* and *M. abscessus*. As demonstrated, PEP-TORCH achieved a perfect $PW_{sp}$ score of 100 for both species across all replicates when analyzing all identified peptides. Further analysis of the peptides identified across different replicates revealed that ~74% and 75% of peptides were common to all replicates of *M. tuberculosis* and *M. abscessus*, respectively (Fig. 3D). In addition, *M. tuberculosis* and *M. abscessus* showed that about 25% of the peptides were identified in either two or only one replicate. To further explore the capability of PEP-TORCH to determine multiple species infections in a single sample (Fig. 3E), samples of *M. tuberculosis* mixed with *M. abscessus* were prepared in various peptide amount ratios (10:90, 25:75, 50:50, 75:25, and 90:10) to simulate co-infection scenarios. These simulated subsets were also analyzed in three replicates to assess the reproducibility of the pipeline's ability to identify mixed samples. The results indicated that PEP-TORCH successfully identified mixed species in these samples, and $PW_{sp}$ scores of the mixed samples were found to be highly correlated with the peptide amount ratios, the Pearson's correlation coefficient (r) of 0.98–0.99 (Fig. 3E).

Therefore, the PEP-TORCH algorithm enhances mycobacterial classification by accurately identifying species and subspecies with high precision and consistency. It is effective in detecting mixed-species infections, marking a notable advancement in mycobacterial detection.

## PEP-TORCH's automated mycobacteria IDs match those from multiple clinical assays

The classification performance of PEP-TORCH was evaluated by analyzing CFP data from 81 samples, which were concurrently assessed using established clinical methods such as MALDI-TOF,

Accuprobe, and/or WGS (Fig. 4A; Table 2). These clinical methods involved further sub-culturing of colonies collected from MGIT samples. Our method saved time on the further solid-culturing days with 13 days average (range from 1 to 39 days) rather than the MALDI-TOF method among all species (Fig. 4B). The results obtained from PEP-TORCH exhibited complete concordance with the outcomes of these clinical approaches, accurately identifying both species and subspecies levels (Table 2).

All samples identified as *M. abscessus*-positive through clinical assessments by MALDI-TOF were consistently classified as *M. abscessus*-positive by PEP-TORCH analysis. A significant portion (87%, 26/30) of these samples exhibited $PW_{sp}$ scores reflecting peptide matches exclusively with *M. abscessus*, yielding a perfect $PW_{sp}$ of 100. The remaining samples presented high *M. abscessus* $PW_{sp}$ scores, ranging from 94.3 to 99.3 (mean 97.3 ± 2.0), slightly lowered due to sporadic peptide matches with species within the *M. abscessus* complex species (Bajaj et al, 2022), which gave minor $PW_{sp}$ scores between 0.4 to 5.7 (Dataset EV3A). Additionally, the PEP-TORCH methodology could differentiate two *M. abscessus* subspecies, *M. abscessus* subsp. *massiliense* and *M. abscessus* subsp. *abscessus*. In total, the PEP-TORCH results classified all the samples at the subspecies level (Fig. 4C; Dataset EV3B), among which 12 (MAB 1- MAB 12) were subjected to confirmation through WGS. Remarkable 100% agreement between PEP-TORCH and WGS sequencing was observed in all instances of *M. abscessus* subsp. *massiliense* (samples MAB 2–5) and *M. abscessus* subsp. *abscessus* identification (samples MAB 6-12) with a perfect $PW_{subsp}$ score of 100 for 11 samples and one sample yielding $PW_{subsp}$ of 99.8 for *M. abscessus* subsp. *abscessus*. Notable cases of "hybrid isolate" were also detected by PEP-TORCH. Hybrid isolate is produced naturally by the lateral transfer of the hybrid *rpoB* gene from *M. abscessus* subsp. *abscessus*, thereby providing the risk of misidentification when *rpoB*-based methods are used in mycobacterial diagnosis (Kim et al, 2019). Therefore, as a standard practice in the current clinical setup, a multilocus sequencing approach is required to identify the two *M. abscessus* subspecies which is generally initiated by *secA* gene sequencing followed by sequencing of additional gene targets (Zelazny et al, 2009). On the contrary, PEP-TORCH streamlines this process by directly identifying specific peptides within the sample, thereby potentially obviating the need for downstream validation techniques to ascertain subspecies information. One case involving the "hybrid species" isolate (MAB 1), as confirmed by WGS yielded $PW_{subsp}$ scores of 62.5 and 37.5 for *M. abscessus* subsp. *abscessus* and *M. abscessus* subsp. *massiliense* respectively. This is in addition to several other unconfirmed hybrid cases (MAB 16, 17, 18, 21, 22, 25, and 27), which were also detected by PEP-TORCH as "hybrid".

Furthermore, *M. avium*-positive were identified with high $PW_{sp}$ scores (92.9 ± 14.3), with six revealing perfect *M. avium* matches (Fig. 4D; Dataset EV3C) and 1 sample with $PW_{sp}$ of 71 for *M. avium* and scores of 19 and 9 for *M. vulneris* and *M. marseillense*, both of which belong to *M. avium* complex. Also, all *M. avium* could be identified as *M. avium* subsp. *hominissuis* with 100 $PW_{subsp}$ score (Dataset EV3D). Five *M. intracellulare* samples were judged *M. intracellulare*-positive by their strong consensus *M. intracellulare* $PW_{sp}$ scores perfectly, with three further identified as *M. intracellulare* subsp. *chimera* isolates. (Fig. 4E; Dataset EV3E,F) Although this observation was based on only 5 *M. intracellulare* samples, it still represented an improvement over current clinical

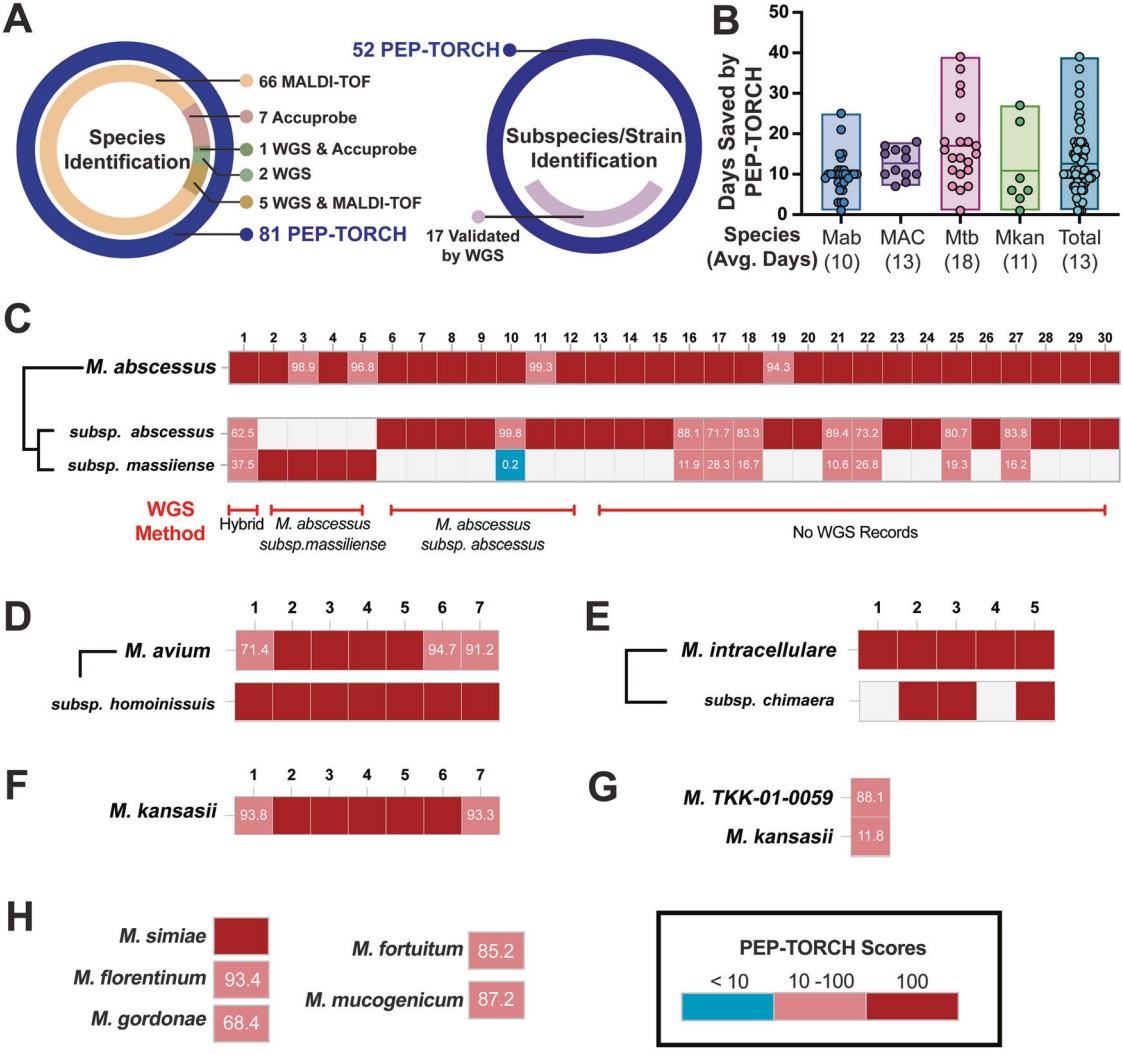

**Figure 4. PEP-TORCH-derived NTM species and subspecies identifications.**

(A) Overall summary of sample identification by PEP-TORCH and other clinical tools at species and subspecies levels. Samples were identified using MALDI-TOF MS, Accuprobe PCR, or WGS. PEP-TORCH provided perfect matches to all these clinical methods. (B) Solid culture time saved by the PEP-TORCH method compared to MALDI-TOF for different species. Species abbreviations: *M. abscessus* (Mab, $n = 29$), *M. avium* complex (MAC, $n = 12$), *M. tuberculosis* (Mtb, $n = 21$), and *M. kansasii* (Mkan, $n = 7$). Data are presented as a floating bar plot representing all results from minimum to maximum values, with the horizontal line inside indicating the mean and all samples as dots inside the bar. (C–H) The PEP-TORCH methodology assigned the peptide weightage (PW) scores for species ($PW_{sp}$) to facilitate identification. All samples (C) attributed to *M. abscessus* by PEP-TORCH $PW_{sp}$ scores were subsequently corroborated as *M. abscessus* through MALDI-TOF MS analysis. Furthermore, subspecies PW scores ($PW_{subsp}$) were assigned to these samples, distinguishing them as either *M. abscessus* subspecies *abscessus* or *M. abscessus* subspecies *massiliense*. Notably, PEP-TORCH exhibited the capability to discern subspecies identities for all samples, however, only 17 samples were verifiably confirmed through WGS analysis, owing to the intricate and resource-intensive nature of WGS procedures. The PEP-TORCH analysis yielded $PW_{sp}$ scores for samples identified as belonging to various species: (D) *M. avium* with $PW_{subsp}$, and (E) *M. intracellulare* with $PW_{subsp}$. (F) *M. kansasii*, (G) one case was diagnosed as co-infected by *M. kansasii / M. avium* (TKK-01-0059, a strain of *M. avium*, from the PEP-TORCH results), and (H) less common species including *M. simiae*, *M. florentinum*, *M. gordonae*, *M. fortuitum*, and *M. mucogenicum*. Heatmaps indicate differences in PW scores with dark red at 100, light red between 10 and 100, and blue was less than 10.

MALDI-TOF results, which cannot distinguish *M. intracellulare* and *M. chimaera* isolates. Samples identified as *M. kansasii*-positive were based on a multifaceted approach, including a combination of MALDI-TOF, Accuprobe and WGS data in the clinical setup. PEP-TORCH exhibited 100% agreement in all the samples identified as *M. kansasii*. Out of the 7 samples examined, 5 displayed a $PW_{sp}$ of 100 for *M. kansasii*, whereas the remaining 2 samples exhibited a $PW_{sp}$ of 93. This slight discrepancy in $PW_{sp}$ values for the latter 2 samples was primarily due to the presence of peptides originating

from species within the *M. kansasii* complex, notably *M. innocens*. Consequently, the collective mean $PW_{sp}$ for the sample set identified as *M. kansasii* was calculated to be 98.1 ± 3 (Fig. 4F; Dataset EV3G). The detection of mixed infections involving *M. innocens* was facilitated by the identification of five combinations of peptides, which would likely have gone undetected by current standard methods due to their low copy numbers. Notably, a sample classified as a co-infected *M. kansasii / M. avium* sample (co-inf 1) identified by MALDI-TOF results had $PW_{sp}$ of 88.1 and

**Table 2.  Comparison of PEP-TORCH results with reference identification.**

| ID reference | Species IDs | | | Subspecies IDs | | |
|---|---|---|---|---|---|---|
| | MALDI (Sequencing)* | PEP-TORCH | Agreement | WGS | PEP-TORCH | Agreement |
| M. tuberculosis Complex | | | | | | |
| *M. tuberculosis* | 24/26 (2/26)* | 26/26 | 100% | | | |
| *M. bovis* | | | | 2/2 | 2/2 | 100% |
| *M. bovis BCG* | | | | 1/1 | 1/1 | 100% |
| *M. africanum* | | | | 2/2 | 2/2 | 100% |
| NTM | | | | | | |
| *M. abscessus* | 30/30 | 30/30 | 100% | | | |
| *M. abscessus* subsp. *abscessus* | | | | 7/7 | 7/7 | 100% |
| *M. abscessus* subsp. *massiliense* | | | | 5/5 | 5/5 | 100% |
| *M. kansasii* | 1/7 (6/7)* | 7/7 | 100% | | | |
| *M. avium* | 7/7 | 7/7 | 100% | | | |
| *M. intracellulare/Chimaera* | 5/5 | 5/5 | 100% | | | |
| Co-infection | 1/1 | 1/1 | 100% | | | |
| *M. gordonae* | 1/1 | 1/1 | 100% | | | |
| *M. mucogenicum* | 1/1 | 1/1 | 100% | | | |
| *M. fortuitum* | 1/1 | 1/1 | 100% | | | |
| *M. simiae*** | 1/1 | 1/1 | 100% | | | |
| *M. floretinum*** | 1/1 | 1/1 | 100% | | | |
| Total | 81/81 | 81/81 | 100% | 17/17 | 17/17 | 100% |

*Sequencing is performed either by polymerase chain reaction (PCR) or whole-genome sequencing (WGS).
**The isolates were received from The BEI Resources Repository. The catalog numbers were NR4434 (*M. simiae*) and NR49073 (*M. florentinum*).

11.9 for *M. kansasii* and *M. TKK-01-0059*, respectively (Fig. 4G; Dataset EV3H). However, at least one source classifies *M. TKK-01-0059* as belonging to the *M. avium* complex (Mnyambwa et al, 2018), while the National Centre for Biotechnology Information (NCBI) classifies it with the *M. tuberculosis* complex. Additionally, samples from five less common mycobacterial species, including *M. gordonae*, *M. mucogenicum*, *M. fortuitum*, *M. simiae*, and *M. floretinum*, were identified with high $PW_{sp}$ scores using the PEP-TORCH pipeline (Fig. 4H; Dataset EV3I). These diverse NTM isolates further support the capability of our pipeline and demonstrate its potential to extend to additional species.

Finally, as shown in Fig. EV3, among the 26 TB clinical cases analyzed, nine were identified as *M. tuberculosis* based on peptides that specifically matched *M. tuberculosis*. The remaining 17 cases were classified as *M. tuberculosis* complex, as the data could not distinguish *M. tuberculosis* from *M. canettii* or *M. orygis*. Both *M. canettii* and *M. orygis* are part of the *M. tuberculosis* complex and can cause tuberculosis in humans. However, these species are rare causes of human infection within the *M. tuberculosis* complex (Riopel et al, 2024; Somoskovi et al, 2009). Detailed detection annotations are provided in Fig. EV3 and Dataset EV3J. Given the rarity of *M. orygis* and *M. canettii*, these species were considered as *M. tuberculosis* during PWsp score calculations in our analysis. All 26 cases had *M. tuberculosis* $PW_{sp}$ scores > 97, with most (24 of 26) scoring 100 (Fig. EV3; Dataset EV3J). PEP-TORCH results also provided subspecies identifications for 19 of these samples. However, since the Unipept database identifies *M. bovis* and *M.*

*africanum* as "sub-variants" of *M. tuberculosis* and thus uses a subspecies rather than species taxonomy when classifying them, these species were identified as *M. tuberculosis* subspecies in our PEP-TORCH results. This analysis provided putative identifications for 13 *M. africanum* and 3 *M. bovis* subspecies and 2 *M. tuberculosis* strain isolates (*M. erdman* and *M. CDC1551*) with high overall $PW_{subsp}$ scores (98.0 ± 5.4) and the majority (84%) had perfect $PW_{subsp}$ scores (Fig. EV3; Dataset EV3K). Notably, there was complete concordance between 5 samples confirmed by WGS and the results from PEP-TORCH for these samples. Further, only 7 of the *M. tuberculosis* samples did not yield subspecies identifications, and this was due to peptide sequence conservations among strains such as *M. CDC1551*, *M. H37Rv*, and *M. H37Ra*.

In summary, PEP-TORCH proved to be a highly effective tool for identifying both species and subspecies levels using CFP samples, exhibiting complete agreement (100%) with clinical standard methods conducted on further subcultures. Its superiority lies in its ability to offer both species and subspecies information simultaneously to provide timely as well as comprehensive pathogen identification without the need for additional downstream techniques to provide hierarchical information. Of particular note is its capability to differentiate macrolide-sensitive (*M. abscessus* subsp. *massiliense*) and -resistant (*M. abscessus* subsp. *abscessus*) *M. abscessus* isolates, a task reliant on multilocus gene sequencing. Additionally, it accurately identified all *M. tuberculosis* at the species level for samples that were otherwise identified at the *M. tuberculosis* complex level. Furthermore, in a noteworthy

instance of a co-infection sample, PEP-TORCH provided the precise proportion of co-infecting pathogens present in the sample which was previously unattainable with existing methods.

## PEP-TORCH results permit species identification from liquid cultures before reaching a threshold for clinical detection

Parallel reaction monitoring (PRM) was next used to evaluate the biomarker utility of a total of 17 peptide signatures identified by PEP-TORCH for the five target species when they were analyzed by a low-resolution triple quadrupole MS system suitable for use in clinical applications (Brzhozovskiy et al, 2022; Chenau et al, 2014; Park et al, 2023; Wee et al, 2019). To this end, targeted proteomics was applied in a total of 63 samples of which 42 samples were pre-validated by PEP-TORCH and 21 samples were newly introduced exclusively for PRM (Table 3).

Peptides were chosen based on two criteria. First, peptides were selected by their detection frequency in all samples judged positive for a given species or complex by PEP-TORCH and clinical data, and by their relative frequency in peptide combinations specific to these groups (Fig. 5A). This was done to select peptides with robust species-specific expression and sufficient combinations to produce robust PW scores. This approach identified six *M. abscessus*, four *M. kansasii*, and seven *M. tuberculosis* peptides that contributed to the five most frequent peptide combinations for each species. *M. avium* and *M. intracellulare* peptides were not included in this analysis since they derived from relatively few samples and yielded lower numbers of species-specific peptides than the *M. abscessus*, *M. kansasii*, and *M. tuberculosis* samples. *M. avium* and *M. intracellulare* peptides selected for further analysis were, therefore, instead chosen by their ranked detection frequency in the list of unique peptides specific for each of these species.

Next, *M. abscessus*, *M. kansasii*, and *M. tuberculosis* peptides that met these criteria were screened to identify peptide combinations that produced the highest median DDA peak area values and peptide-spectrum match (PSM) scores to identify peptide pairs that had strong MS signal intensities (Fig. EV4A) and confident database identifications expected to allow their consistent detection in MGIT samples. This analysis identified three peptides for *M. tuberculosis*, *M. abscessus*, and *M. kansasii* that had the highest average combined scores in samples for each of these species (Fig. 5B). In addition to 3 peptides in *M. tuberculosis* samples, six

more peptides detected by DDA that were also matched to *M. tuberculosis* and had high PSM scores ($-\log_{10} P > 55$), were also tested by PRM in 14 randomly chosen *M. tuberculosis*-positive MGIT cultures (Fig. EV4B). *M. avium* and *M. intracellulare* peptides were analyzed similarly to identify single peptides that had the highest combined DDA peak area values and PSM scores, and this approach identified 3 *M. avium*- and 2 *M. intracellulare*-specific target peptides.

The four selected *M. tuberculosis* peptides were derived from two secreted proteins that play important roles in virulence: ESAT-6 (LAAAWGGSGSEAYQGVQQK, WDATATELNNALQNLAR) and CFP-10 (TQIDQVESTAGSLQGQWR and QKLDEISTNIR). Similarly, the *M. kansasii* peptides were mapped to the *M. kansasii* orthologs of ESAT-6 (WDATAQELNNALQNLSR) and CFP-10 (TQIDQVES TAASLQAQWR and AELEEISTNIR). Peptides from ESAT-6 and CFP-10 in *M. kansasii* and *M. tuberculosis*, although mapped to the same sites in the corresponding proteins, the peptides differed at only three and two amino acid positions, respectively, suggesting the power of the MS technology to discriminate between two closely matched peptides to identify two different species (Fig. EV2). *M. abscessus*, *M. avium*, and *M. intracellulare* do not express these virulence factors. However, similar example of discrimination of closely matched peptides could also be observed in *M. intracellular* and *M. avium* where *M. intracellulare*-specific peptide (NYSENFYAPQADPLW LAWPNHMK) and *M. avium*-specific peptide (AHWFYALSPQDR) mapped to a protein with unknown function, DUF5078 domain-containing protein that is highly conserved between these two species (95.9% identity) but differs at a single position in these two matching *M. avium* and *M. intracellulare* peptides (Fig. EV5A). The three remaining *M. avium*-specific peptides, HPDLHQQLQQR and AAGA-GATVLNVSK, were mapped to haemophore heme-binding domain-containing protein, putative thiol peroxidase, and diacylglycerol acyltransferase respectively. Peptides from low molecular weight T-cell antigen (VSMINQVK, GNQGIEYVIPVFQQMVR), secreted protein (TYLDGQPAAK, AQTSGNPLLTSLLN), and proline-rich antigen-like protein (LIGEFDTNEAVSHGPVEVK) were used as targeted peptides to discriminate *M. abscessus* from other mycobacteria using species-specific peptides (Fig. EV5B).

These PRM peptide targets were used to identify mycobacteria species present in all the MGIT samples with sufficient material remaining for a second LC-MS/MS analysis. Fourteen *M. tuberculosis* complex samples had sufficient material for this analysis, but few *M. abscessus*, *M. kansasii*, and *M. avium* samples had sufficient material and new samples were therefore added to these groups (Fig. 5C; Dataset EV4). All four *M. tuberculosis* peptides were detected in all *M. tuberculosis*-positive samples, except for one sample from a child who developed a rare, disseminated infection (BCGosis) after receiving the *M. bovis* BCG vaccine. This vaccine, derived from an attenuated *M. bovis* strain that lacks CFP-10 and ESAT-6, typically does not cause the detection of *M. tuberculosis* peptides in vaccinated individuals, unless they are later infected with *M. tuberculosis* or, in rare cases, develop BCGosis.

Next, we evaluated if our approach could also identify specific bacteria in MGIT cultures before mycobacterial growth was detectable. Automated MGIT systems need high bacterial biomass ($10^5$ CFU/ml) to detect mycobacterial growth, and thus can require several weeks to detect a positive result and additional time for subcultures used for species identification. Since LC-MS/MS was

**Table 3.  Peptide targets validated in mycobacterial species by parallel reaction monitoring mass spectrometry.**

| ID reference | Samples | | | | |
| | Sample sub-set from Table 2 | New sample set | Total | Validation by PRM | Agreement |
|---|---|---|---|---|---|
| *M. tuberculosis* | 14/14 | 0 | 14 | 14/14 | 100% |
| *M. abscessus* | 16/16 | 3/3 | 19 | 19/19 | 100% |
| *M. kansasii* | 6/6 | 1/1 | 7 | 7/7 | 100% |
| *M. avium* | 3/3 | 11/11 | 14 | 14/14 | 100% |
| *M. intracellulare* | 3/3 | 6/6 | 9 | 9/9 | 100% |
| Total | 42 | 21 | 63 | 63/63 | 100% |

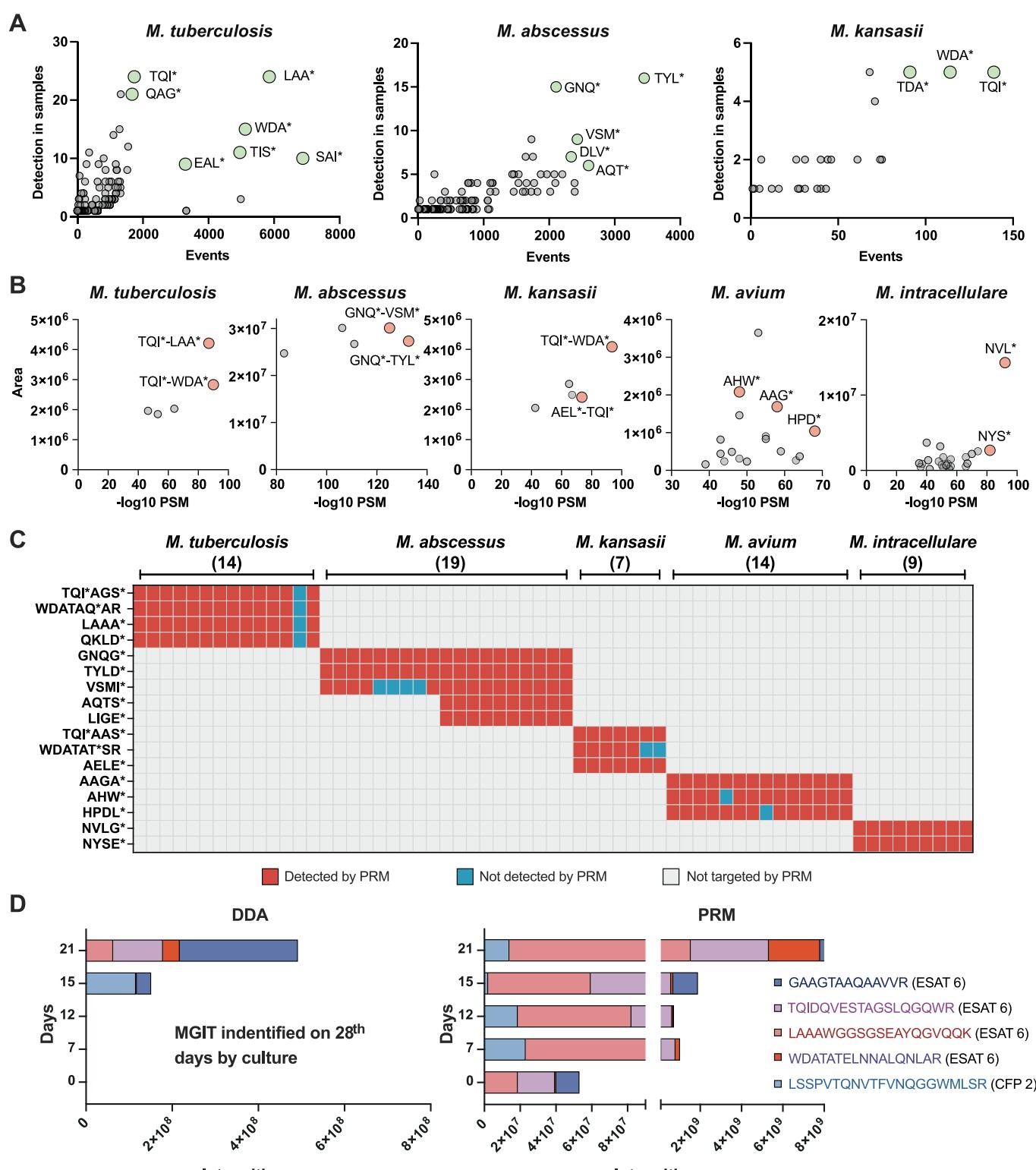

found to robustly distinguish MGIT cultures at the first sign of culture positivity, we therefore analyzed CFP samples collected at successive intervals (day 7, 12, 15, and 21) from parallel MGIT samples generated from an *M. tuberculosis* isolate that was growth-positive at day 28 by both DDA and PRM assay. This analysis

employed all nine *M. tuberculosis* unique peptides that were reproducibly detected by PRM in 14 *M. tuberculosis*-positive MGIT cultures (Fig. EV4B).

Of the 9 unique peptides, DDA results identified 5 in the serial MGIT culture samples, with 3 target peptides detected on day 15

**Figure 5. Diagnostic sensitivity of PRM target peptides and earlier *M. tuberculosis* detection.**

(A) Peptides utilized for the identification of *M. tuberculosis*, *M. abscessus*, and *M. kansasii* were ranked based on their occurrence frequency across samples and their propensity to combine with other peptides to form unique combinations indicative of specific species identification. Peptides, highlighted as green bubbles, were selected based on their higher frequency than other peptides in identifying a particular species. This analysis could not be performed in *M. avium* and *M. intracellulare* owing to the low number of unique peptide combinations formed. (B) Subsequently, the peptides identified in (A) were systematically paired with each other and ranked according to their peak area and $-\log_{10}$ peptide-spectrum match (PSM) scores to determine the most confidently identified peptides, highlighted as red bubbles. These selected peptide pairs were then designated as targeted peptide sequences for parallel reaction monitoring (PRM) assays specific to each species. (C) The frequency of detection of the PRM target peptides in culture filtrate samples containing the respective indicated species is depicted, with positive and negative detection results denoted by red and blue boxes, respectively. (D) Comparative analysis of the peak intensities obtained through data-dependent acquisition (DDA) and PRM for the selected *M. tuberculosis* target peptides is presented at various time points post-culture inoculation. Notably, in a clinical setup, the BACTEC MGIT 960 instrument confirmed the positivity of the culture on day 28 post-inoculation. However, our peptide analysis demonstrated that PRM targeting specific peptides could detect *M. tuberculosis* infection as early as the 7th and 12th days post-inoculation. Our peptidome analysis using the DDA method also could identify *M. tuberculosis* infection by the 15th day, providing evidence of both DDA and PRM's enhanced sensitivity and the potential for accelerated detection timelines, thus facilitating earlier intervention and treatment initiation compared to the current clinical setup. Source data are available online for this figure.

and 4 peptides on day 21, and all of these peptides were derived from ESAT-6 except for 1 originating from CFP-2 (Fig. 5D). PRM results for these samples detected 4 target peptides at days 7 and 12, and in addition to another target peptide from CFP-2 on day 15 and day 21. Both the DDA and PRM results indicated that the number and relative expression of detected peptides progressively increased with time. Although peptides derived from the same protein exhibited substantial signal intensity differences, not all of them consistently increased with culture time.

In summary, our approach not only validated target peptides to be readily used in clinical settings but also showed that signature peptides could be detected at least 7–12 days earlier in the MGIT samples before the MGIT turned clinically positive for mycobacterial growth. This shows the potential of our peptidomics approach could provide early detection of mycobacterial infections, allow for prompt intervention, and improve patient outcomes.

# Discussion

The identification of mycobacteria species and subspecies is frequently a complex and slow process that requires the use of multiple diagnostic protocols. Our results, however, indicate that integration of a streamlined LC-MS/MS method with the automated PEP-TORCH algorithm analysis approach permits rapid and precise simultaneous identification of mycobacteria at species and subspecies resolution through the detection of specific peptides and peptide combinations. The rapid and precise diagnoses provided by PEP-TORCH thus have significant potential to improve patient outcomes by facilitating early and accurate diagnosis to permit more effective treatment interventions.

In contrast to previous studies which analyze the limited peptide markers to identify species (Chen et al, 2019), our pipeline automated the search for peptides that could distinguish between different taxa levels and provide scores that could be used to assist clinicians in diagnostic and therapeutic intervention decisions. In addition, PEP-TORCH has several advantages over conventional mycobacteria identification methods since it can rapidly identify mycobacteria isolates to the subspecies level with a single LC-MS/MS analysis. It does so by comparing detected biomarker peptides against protein database data to detect even single position differences in the peptide sequence of conserved proteins. This is an advantage over methods that use MS spectra data for identifications since spectra differences, but not peptide

identifications, can be affected by culture conditions. In this study, we demonstrated that PEP-TORCH $PW_{sp}$ and $PW_{subsp}$ scores can accurately identify species and subspecies in supernatants of MGIT cultures, including co-infections. Moreover, our data on detecting peptides before the MGIT turned positive also showed how DDA and PRM data both permit more rapid identifications than obtained from MALDI-TOF or WGS analyses of subcultures of growth-positive MGIT samples used to confirm the identity of a mycobacterial species or its complex. This expedited identification process can enable healthcare providers to make more informed and timely treatment decisions, leading to improved patient outcomes, especially in cases where prompt intervention is critical, such as in neonatal or immunocompromised patient populations.

Rapid and accurate identification of subspecies is important for appropriate treatment initiation, as some subspecies have different inherent drug resistance or host interactions (Manca et al, 1999). MALDI-TOF analyses have limited ability to distinguish subspecies of *M. abscessus* and closely related species such as *M. intracellulare* and *M. chimaera* (Toney et al, 2022), since they rely on detecting specific spectral patterns rather than specific proteins or peptides. Their spectral libraries also may lack sufficient coverage to identify less common species or variants of more common species, problems that are less likely to affect LC-MS/MS that can discriminate samples based on specific peptide differences (Brown-Elliott et al, 2019; Buckwalter et al, 2016). Moreover, it's worth noting that ~45–70% of *M. abscessus* subsp. *abscessus* strains exhibit inducible resistance to macrolide drugs like clarithromycin and azithromycin after 14 days of incubation, while almost 81% of *M. abscessus* subsp. *massiliense* strains remain drug-susceptible (Guo et al, 2021). Given that macrolides are considered the cornerstone of treatment for *M. abscessus* infections, accurate subspecies identification becomes imperative. Our identification of a case with a hybrid subspecies underscores the potential of PEP-TORCH to identify both drug-sensitive and drug-resistant subspecies. However, a larger sample size is necessary to further efficacy validation in clinical settings. Moreover, current clinical diagnostics can identify *M. tuberculosis* as a member of the *M. tuberculosis* complex, unlike PEP-TORCH results, which distinguished BCGosis from zoonotic TB caused by *M. bovis* and both from infections caused by other *M. tuberculosis* complex species and strains. This is noteworthy since *M. bovis* exhibits inherent pyrazinamide resistance to affect treatment decisions.

Most conventional methods first identify species and then identify subspecies. Only very limited methods could identify both

levels simultaneously. In addition, most clinical standard methods lack the identification of multiple infections. Recently, line probe assays consisting of PCR and Southern blot hybridization, such as FluorType assay could provide co-infection information (Luukinen et al, 2024) however with limited detection ability for *M. intracellulare* and *M. chimaera* (42%). It is important to note that PEP-TORCH's identifications of co-infection in addition to single infections were consistent with clinical diagnoses obtained through various other clinical methods. Moreover, as evident from the simulated mixed culture samples, PEP-TORCH not only determines co-infections but also reliably determines the proportion of co-infection involving multiple species in a single sample through its $PW_{sp}$ scores. This insight is crucial as it indicates the dominance of one species over another, aiding in the formulation of precise treatment strategies. PEP-TORCH eliminates the need for extended culture times (averaging 13 days in our validation set), significantly accelerating diagnosis and treatment. It is being developed into a user-friendly, one-click software, streamlining data processing and reducing the need for experienced technicians, which facilitates seamless integration into clinical laboratories. While TB remains a challenge in resource-limited settings, the rise in NTM infections in developed countries underscores the relevance of PEP-TORCH for centralized labs in these regions. Our targeted mass spectrometry approach, leveraging peptidomics, aligns well with clinical labs already equipped with triple quadrupole mass spectrometry. Although mass spectrometry requires an initial capital investment, the low per-sample reagent cost ensures cost-effectiveness for routine use. The simple protocol supports high-throughput applications, requiring minimal training and resources, further enhancing its scalability and suitability for clinical adoption.

A limitation of this study was the unavailability of WGS data for all specimens, which restricted our capacity to validate PEP-TORCH results concerning subspecies identification and co-infection. For instance, in the case of sample MKA3, both PEP-TORCH and Accuprobe concurred on its identification as *M. kansasii*, but PEP-TORCH also generated a low *M. tuberculosis* score that could not be cross-checked with WGS data. Another limitation of our PRM test is that the peptides selected for *M. kansasii* and *M. intracellulare* do not uniquely match only these species (Dataset EV4). However, the additional species matched—such as *M. gastri*, *M. persicum*, and *M. timonense*—are rarely associated with human disease. Given the rarity of these species, the targeted results predominantly represent *M. kansasii* and *M. intracellulare*. The pipeline is designed to detect a broader range of human-infecting NTM species. While this study primarily tested high-prevalence species for validation, we also included five less common species. Future research will focus on further validation with additional less common species. Also, with the integrative applications of PEP-TORCH with other mass spectrometry-driving methods of mycobacteria-specific or -associated metabolites (Chen et al, 2022), or other cutting-edge biomarkers (Das et al, 2024; MacLean et al, 2019), we believe that it would open a new avenue for diagnosis with higher specificity and accuracy for mycobacterial infections with additional less common species or complex infection settings, such as HIV/AIDS or *M. tuberculosis*/NTM co-infections. Our validation cohort, consisting of 102 clinical samples, may not be particularly large in scale, but it retains its significance as a valuable resource. Significantly, it contributes to our comprehension of a substantial portion of the annual NTM cases in the United States, estimated at 4.7 cases per 100,000 person-years (Winthrop et al, 2020).

This is a pathogen deserving of increased research attention, as its incidence is rising globally, and the development of diagnostic technologies for it lags behind. We hope that our study not only provides a novel detection platform but also raises awareness for further research targeting this pathogen.

In summary, this study highlights the ability of peptidomics data to rapidly and specifically identify mycobacterial samples, particularly among closely related species and subspecies. This should enhance the accuracy of species identifications within the mycobacterium superfamily to benefit clinical diagnosis. Further, the automated algorithm-based pipeline can be employed as a universal means to analyze complex MS data from clinical isolates as long as these isolates are represented in the MS databases used to evaluate the specificity of the detected peptides.

# Methods

### Reagents and tools table

| Reagent/resource | Reference or source | Identifier or catalog number |
|---|---|---|
| **Experimental models** | | |
| Clinical MGIT-filtered supernatant samples | National Institutes of Health (NIH) Clinical Center, Department of Laboratory Medicine | Dataset EV1 |
| Mycobacterium strain - *M. simiae* | Bei Resources | NR4434 |
| Mycobacterium strain - *M. florentinum* | Bei Resources | NR49073 |
| **Chemicals, enzymes, and other reagents** | | |
| Acetonitrile | Fisher Chemical | #A998-4 |
| Acetone | Fisher Chemical | #A18-20 |
| Methanol | Fisher Chemical | #A412-4 |
| Trifluoroacetic acid | Fisher Chemical | #A116-50 |
| Isopropanol | Fisher Chemical | #A464SK-4 |
| Dithiothreitol | Thermo Scientific Chemicals | #PI20290 |
| Indole-3-acetic acid | Thermo Scientific Chemicals | #AAA1055614 |
| Ammonium bicarbonate | Thermo Scientific Chemicals | #AC393212500 |
| Trypsin | Promega | #PRV5117 |
| **Software** | | |
| Graphpad Prism Version 10.4.1 (532). | https://www.graphpad.com/features | GraphPad by Dotmatics |
| Rstudio 2022.12.0 build 353 | https://posit.co/download/rstudio-desktop/ | Posit Open-Source Data Science Company |
| BSI PEAKS Studio 10.6 Pro build 20201221 | https://www.bioinfor.com/peaks-xpro/ | BSI Bioinformatics Solution Inc. |
| Skyline 23.1.0 | https://skyline.ms/project/home/begin.view | MacCoss Lab Software |

## Sample collection and clinical analyses

MGIT CFP samples analyzed in this study were obtained from clinical patients enrolled in IRB-approved protocols at the National Institutes of Health Clinical Center (Dataset EV1). Informed consent was obtained from all human subjects and confirmed that the experiments conformed to the principles set out in the WMA Declaration of Helsinki and the Department of Health and Human Services Belmont Report. All clinical isolates were inoculated into MGIT tubes (Remel; Lenexa, KS) containing Middlebrook 7H9, OADC, and PANTA growth supplements. In addition, two mycobacterial strains, *M. simiae* (NR4434) and *M. floretinum* (NR49073), were acquired from BEI Resources, NIAID, NIH, and were cultured using the same protocol as the clinical samples.

Clinical samples were decontaminated using the N-acetyl-L-cysteine/sodium hydroxide method. Samples were processed for AFB smear with auramine-rhodamine (Becton Dickinson, Sparks, MD), and AFB culture including Middlebrook 7H11 plates and MGIT tubes. Following inoculation, MGIT tubes were incubated in Bactec 960 (BD BACTEC™ MGIT™). MGIT-positive cultures are further on solid media, and mycobacteria were identified at species or subspecies level by MALDI-TOF MS and if needed targeted sequencing. MGIT CFP samples used for proteomics analysis were generated by passing 3 mL of all growth-positive MGIT cultures through 0.22 μm filters and storing the resulting filtrates at −80 °C until use.

## CFP sample fractionation

CFP samples were rapidly thawed to room temperature and aliquoted for subsequent analyses. One 100 μl aliquot of each sample was supplemented with an equal volume of acetonitrile and centrifuged at $16,000 \times g$ for 25 min at 15 °C. Supernatants were collected and precipitated overnight with four times the volume of acetone overnight then centrifuged at $10,000 \times g$ for 10 min at 4 °C. The resulting pellets were re-dissolved in 50 mM ammonium bicarbonate. Protein concentrations were determined using a BCA protein assay reagent kit (Thermo Scientific, Rockford, Illinois, USA). SDS-PAGE gels (4–20% Mini-PROTEAN TGX Precast Protein Gels, BioRad, USA) were loaded with 10 μg aliquots of the supernatant and resuspended pellet material of each sample, subjected to electrophoresis, stained with Coomassie blue, imaged with a BioRad ChemiDoc Imaging System, and these images were processed and analyzed with ImageLab software (BioRad).

## Trypsin digestion

A 50 μg aliquot of the acetonitrile-fractionation supernatant of each CFP sample was then reduced in 20 mM dithiothreitol for 10 min at 91 °C with mixing and then sonicated for 5 min. These samples were then adjusted to 25 mM iodoacetamide and alkylated for 20 min in the dark, before overnight digestion overnight at 37 °C using 1 μg of sequencing-grade trypsin (Promega, USA). The resulting peptide samples were then acidified by the addition of trifluoroacetic acid (TFA) to a 0.1% final concentration. Styrene-divinylbenzene reverse phase sulfonate (SDB-RPS) stage tips used to process 20 μg aliquots of these samples were then prepared by packing SDB-RPS discs into 200 μl pipette tips with an 18-gauge

syringe needle. Each stage tip was wetted by the addition of 50 μl of 100% acetonitrile and then centrifuged at $1000 \times g$ for 1 min, then equilibrated with 50 μl of 30% methanol/1% TFA and centrifuged at $1000 \times g$ for 3 min. Equilibrated stage tips were then loaded with 20 μg tryptic peptide aliquots and washed with 100 μl of a 99% propanol/0.1% TFA solution using a $1000 \times g$ for 2 min centrifugation step, after which peptides were eluted with 60 μl of elution buffer (80% acetonitrile, 1% ammonium hydroxide), dried, and reconstituted in 2% acetonitrile and 0.1% formic acid and subjected to DDA or PRM mode LC-MS/MS analysis for data acquisition.

## LC-MS/MS analysis

For the DDA analyses, peptide samples were analyzed using an Ultimate 3000 nanoLC system (Thermo Fisher Scientific) coupled via a nano-electrospray ion source (Thermo Fisher Scientific) to a Thermo Orbitrap mass spectrometer (Thermo Fisher Scientific). Samples were loaded onto a 100 μm I.D. × 2.5 cm, C18 trap column and a PepMap RSLC C18 (2 μm ID, 75 μm × 25 cm) analytical column in buffer A (0.1% formic acid in water). Fractions were eluted with a linear gradient of 5% to 23% buffer B (80% acetonitrile in 0.1% formic acid) over 100 min. Following linear separation, the column was adjusted to 40% buffer B over 16 min and finally stepped to 98% buffer B for 4 min, then re-equilibrated to 5% buffer B prior to the next injection. After adjusting each sample to an estimated 0.5 to 1.0 μg on a column, samples were analyzed in DDA topN = 30 with 30 s dynamic exclusion. Precursor spectra were collected from 400 to 1600 *m/z* at 60,000 resolutions (automated gain control (AGC) target of 5e5, max IT of 50 ms). MS/MS data were collected on +2H to +6H precursors achieving a minimum AGC of 8e3. MS/MS scans were collected at 15,000 resolutions (AGC target of 5e5, max IT of 80 ms) with an isolation width of 1.6 *m/z* with a normalized collision energy (NCE) of 30.

For the PRM analyses, samples were loaded onto a PepMap RSLC C18 column (3 μm ID, 75 μm × 15 cm) at a 300 nl/min flow rate in Buffer A. Samples were eluted with a linear gradient of 2% to 37% buffer B over 21 min, and then the column was adjusted to 60% buffer B over 10 min, stepped to 99% buffer B for 6 min, and then re-equilibrated to 2% buffer B prior to the next sample injection. The MS1 had a resolution of 30,000 and a mass range between 150 and 2000 *m/z*; The maximum ion injection time was 200 ms with an AGC target of 3e6. The MS/MS had a resolution of 15,000 and a maximum injection time of 100 ms with an AGC target of 2e5; the isolation window was 1.2 *m/z*. The NCE was set at 28–32 depending on the length of the target peptide sequence. All the peptide information targeted in the study is presented in Dataset EV2.

## Peptide data analysis

Raw MS datasets were analyzed using the PEAKS DB search in PEAKS Studio 10.5 software (Bioinformatics Solutions Inc., Waterloo, Canada). Peptide sequences were searched against a protein database targeting UniProt taxon ID: 1762 (Mycobacteriaceae), which included 2,084,438 entries from 7108 descendant taxonomic groups. This database was downloaded from UniProtKB on August 16, 2022. The parameters of PEAKS are as follows: precursor tolerance is 20 ppm, fragment tolerance is 0.02 Da, two missed tryptic cleavages were

allowed, and dynamic modification is the oxidation of Met. The PRM data were imported into Skyline for analysis.

## Taxonomy analysis

High-confidence peptides ($-\log_{10} P > 35$) were searched against Unipept (https://unipept.ugent.be/) API code with the developed automated pipeline in R Studio to provide species identification. This threshold was decreased to $-\log_{10} P > 20$ for peptides used for subspecies identification. The complete code used in this algorithm is available at: https://github.com/FanLab2019/NTM_PEPTORCH.

Peptide sets of all samples were run through a PEP-TORCH algorithm built in the R Studio 2022.12.0 platform. Peptides were analyzed with Unipept 2.1.3 API code to generate peptide-taxon matrixes that assigned unique peptides and specific taxa to the x- and y-axes of these matrixes. Each matrix entry was then screened to identify peptides that belonged to non-mycobacteria species or mycobacteria that did not derive from a clinical sample, which were then eliminated from the matrix. The PEP-TORCH algorithm then searched these reduced peptide-taxon matrixes to identify peptides that identified specific mycobacteria species and subspecies when they were used as single- or multi-peptide biomarkers in the form of a formula table. The explanation of the score system is in the Appendix Table S2.

## Graphics

Figures 1, 2A–C, and the synopsis graphics were created with BioRender.com.

## Statistical analysis

Experiments were randomized, and investigators were not blinded to allocation during both the experiments and outcome assessment. For single comparisons between two groups with paired samples, a two-tailed paired Student's *t* test was used. Correlation was assessed using Pearson's correlation coefficient. All statistical analyses were performed in GraphPad Prism version 10.4.1. Statistical significance was defined as $P < 0.05$, with exact $P$ values reported for values below this threshold.

## Data availability

The datasets produced in this study are available in the following database: Computer scripts: GitHub (https://github.com/FanLab2019/NTM_PEPTORCH) and Mass spectrometry data: PRIDE PXD059923.

The source data of this paper are collected in the following database record: biostudies:S-SCDT-10_1038-S44321-025-00207-5.

## Peer review information

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

## Acknowledgements

The authors would like to thank all the technologists and staff in the Microbiology Service at the National Institutes of Health (NIH) Clinical Center who helped us in collecting the culture samples. This study was supported by the startup funds and pilot funding from the Tulane COBRE for Clinical and Translational Research in Cardiometabolic Diseases and NIH grants (S10OD032453 and P20GM109036) to JF, NIH grant (R21AI169583) to BN, NIH grants (U01CA214254, R01HD090927, R01AI144168 and R01HD103511) and United States Department of Defense (W8IXWH1910926) to TYH. Support was provided through Jim and Betty Karam postdoctoral scholarship funds, Tulane University to SM. Additional support has been provided by the Division of Intramural Research (DIR) of the Clinical Center, NIH to AZ. The following reagents were obtained through BEI Resources, NIAID, NIH: *Mycobacterium simiae*, Strain MO-323, NR4434 and *Mycobacterium florentinum*, Strain FI-93171T, NR49073.

## Author contributions

**Duran Bao**: Conceptualization; Data curation; Software; Formal analysis; Validation; Investigation; Visualization; Methodology; Writing—original draft; Writing—review and editing. **Sudipa Maity**: Conceptualization; Data curation; Software; Formal analysis; Validation; Visualization; Methodology; Writing—original draft; Writing—review and editing. **Lingpeng Zhan**: Conceptualization; Data curation; Formal analysis; Validation; Investigation; Visualization; Methodology; Writing—original draft. **Seungyeon Seo**: Resources; Methodology; Writing—review and editing. **Qingbo Shu**: Methodology. **Christopher J Lyon**: Visualization; Writing—review and editing. **Bo Ning**: Supervision; Funding acquisition; Visualization; Writing—original draft; Writing—review and editing. **Adrian Zelazny**: Resources; Data curation; Validation; Writing—original draft; Writing—review and editing. **Tony Y Hu**: Conceptualization; Writing—review and editing. **Jia Fan**: Conceptualization; Resources; Data curation; Software; Formal analysis; Supervision; Funding acquisition; Validation; Investigation; Visualization; Methodology; Writing—original draft; Project administration; Writing—review and editing.

Source data underlying figure panels in this paper may have individual authorship assigned. Where available, figure panel/source data authorship is listed in the following database record: biostudies:S-SCDT-10_1038-S44321-025-00207-5.

## Disclosure and competing interests statement

A US patent application has been filed by JF, DB, SM and TYH on the use of the PEP-TORCH algorithm. All other authors declare that they have no competing interests.

# Expanded View Figures

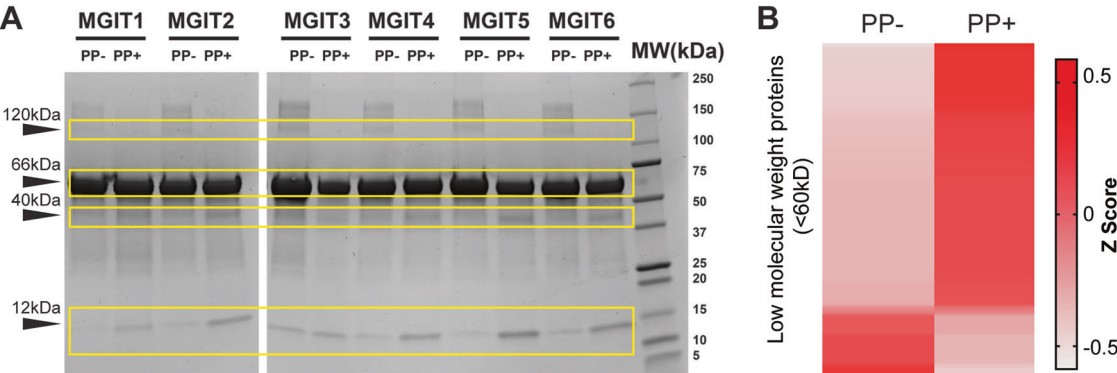

**Figure EV1. Characterization of MGIT CFP samples before and after precipitation.**

(A) SDS-PAGE analysis of protein size distributions in six MGIT CFP supernatant samples before (PP-) and after (PP+) precipitation with 50% acetonitrile, analyzed by gels stained with Coomassie blue. The 120 kDa, 66 kDa, 40 kDa, and 12 kDa protein markers are highlighted. (B) LC-MS/MS analysis of these six CFP samples showed at least a two-fold increase in the number of low molecular weight proteins ( < 60 kDa) with enriched intensities following precipitation.

**A** Species: *M. tuberculosis*; Peptide: WDATA**T**ELN**N**ALQNL**A**R
m/z:951.4816; z = 2; RT: 75.40; -logP = 96.07, ppm = 1.2

Species: *M. kansasii*; Peptide: WDATA**Q**ELN**S**ALQNL**S**R
m/z:959.4832; z = 2; RT: 109.41; -logP = 88.27, ppm = 4.9

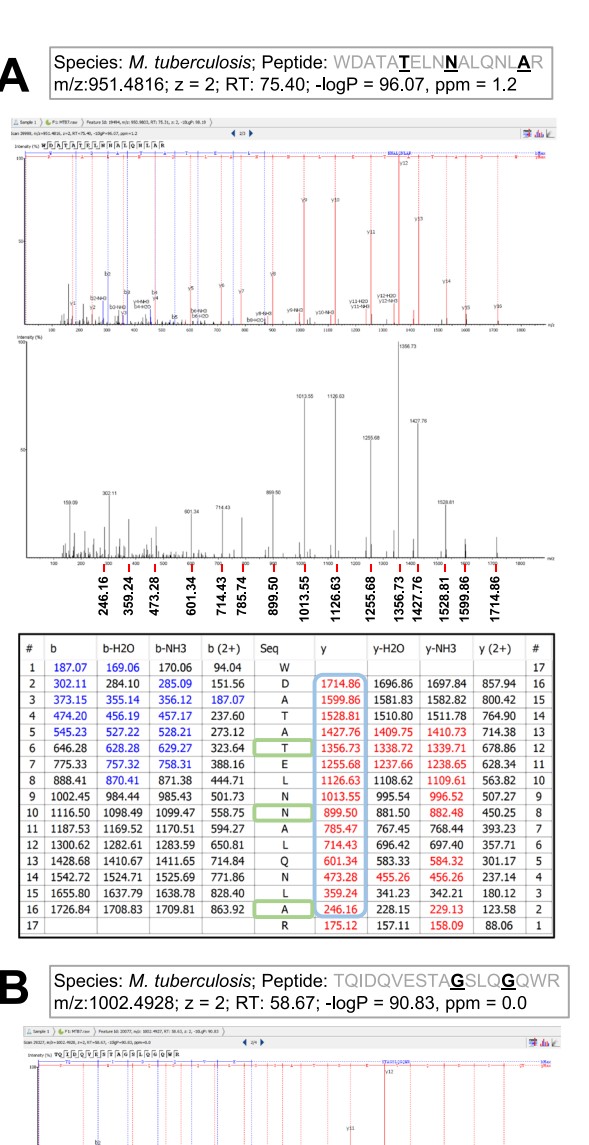
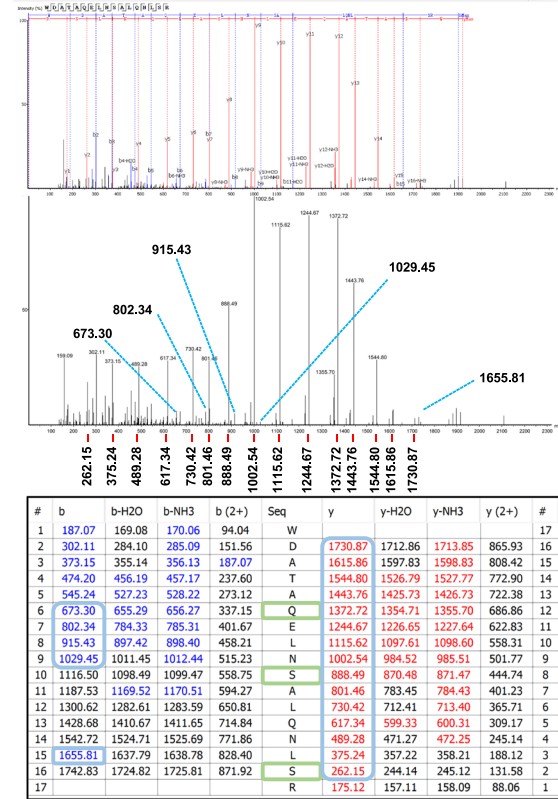

| # | b | b-H2O | b-NH3 | b (2+) | Seq | y | y-H2O | y-NH3 | y (2+) | # |
|---|---|-------|-------|--------|-----|---|-------|-------|--------|---|
| 1 | 187.07 | 169.06 | 170.06 | 94.04 | W | | | | | 17 |
| 2 | 302.11 | 284.10 | 285.09 | 151.56 | D | 1714.86 | 1696.86 | 1697.84 | 857.94 | 16 |
| 3 | 373.15 | 355.14 | 356.12 | 187.07 | A | 1599.86 | 1581.83 | 1582.82 | 800.42 | 15 |
| 4 | 474.20 | 456.19 | 457.17 | 237.60 | T | 1528.81 | 1510.80 | 1511.78 | 764.90 | 14 |
| 5 | 545.23 | 527.22 | 528.21 | 273.12 | A | 1427.76 | 1409.75 | 1410.73 | 714.38 | 13 |
| 6 | 646.28 | 628.28 | 629.27 | 323.64 | T | 1356.73 | 1338.72 | 1339.71 | 678.86 | 12 |
| 7 | 775.33 | 757.32 | 758.31 | 388.16 | E | 1255.68 | 1237.66 | 1238.65 | 628.34 | 11 |
| 8 | 888.41 | 870.41 | 871.38 | 444.71 | L | 1126.63 | 1108.62 | 1109.61 | 563.82 | 10 |
| 9 | 1002.45 | 984.44 | 985.43 | 501.73 | N | 1013.55 | 995.54 | 996.52 | 507.27 | 9 |
| 10 | 1116.50 | 1098.49 | 1099.47 | 558.75 | N | 899.50 | 881.50 | 882.48 | 450.25 | 8 |
| 11 | 1187.53 | 1169.52 | 1170.51 | 594.27 | A | 785.47 | 767.46 | 768.44 | 393.23 | 7 |
| 12 | 1300.62 | 1282.61 | 1283.59 | 650.81 | L | 714.43 | 696.42 | 697.40 | 357.71 | 6 |
| 13 | 1428.68 | 1410.67 | 1411.65 | 714.84 | Q | 601.34 | 583.33 | 584.32 | 301.17 | 5 |
| 14 | 1542.72 | 1524.71 | 1525.69 | 771.86 | N | 473.28 | 455.26 | 456.26 | 237.14 | 4 |
| 15 | 1655.80 | 1637.79 | 1638.78 | 828.40 | L | 359.24 | 341.23 | 342.21 | 180.12 | 3 |
| 16 | 1726.84 | 1708.83 | 1709.81 | 863.92 | A | 246.16 | 228.15 | 229.13 | 123.58 | 2 |
| 17 | | | | | R | 175.12 | 157.11 | 158.09 | 88.06 | 1 |

| # | b | b-H2O | b-NH3 | b (2+) | Seq | y | y-H2O | y-NH3 | y (2+) | # |
|---|---|-------|-------|--------|-----|---|-------|-------|--------|---|
| 1 | 187.07 | 169.08 | 170.06 | 94.04 | W | | | | | 17 |
| 2 | 302.11 | 284.10 | 285.09 | 151.56 | D | 1730.87 | 1712.86 | 1713.85 | 865.93 | 16 |
| 3 | 373.15 | 355.14 | 356.13 | 187.07 | A | 1615.86 | 1597.83 | 1598.83 | 808.42 | 15 |
| 4 | 474.20 | 456.19 | 457.17 | 237.60 | T | 1544.80 | 1526.79 | 1527.77 | 772.90 | 14 |
| 5 | 545.24 | 527.23 | 528.22 | 273.12 | A | 1443.76 | 1425.73 | 1426.73 | 722.38 | 13 |
| 6 | 673.30 | 655.29 | 656.27 | 337.15 | Q | 1372.72 | 1354.71 | 1355.70 | 686.86 | 12 |
| 7 | 802.34 | 784.33 | 785.31 | 401.67 | E | 1244.67 | 1226.65 | 1227.64 | 622.83 | 11 |
| 8 | 915.43 | 897.42 | 898.40 | 458.21 | L | 1115.62 | 1097.61 | 1098.60 | 558.31 | 10 |
| 9 | 1029.45 | 1011.45 | 1012.44 | 515.23 | N | 1002.54 | 984.52 | 985.51 | 501.77 | 9 |
| 10 | 1116.50 | 1098.49 | 1099.47 | 558.75 | S | 888.49 | 870.48 | 871.47 | 444.74 | 8 |
| 11 | 1187.53 | 1169.52 | 1170.51 | 594.27 | A | 801.46 | 783.45 | 784.43 | 401.23 | 7 |
| 12 | 1300.62 | 1282.61 | 1283.59 | 650.81 | L | 730.42 | 712.41 | 713.40 | 365.71 | 6 |
| 13 | 1428.68 | 1410.67 | 1411.65 | 714.84 | Q | 617.34 | 599.33 | 600.31 | 309.17 | 5 |
| 14 | 1542.72 | 1524.71 | 1525.69 | 771.86 | N | 489.28 | 471.27 | 472.25 | 245.14 | 4 |
| 15 | 1655.81 | 1637.79 | 1638.78 | 828.40 | L | 375.24 | 357.22 | 358.21 | 188.12 | 3 |
| 16 | 1742.83 | 1724.82 | 1725.81 | 871.92 | S | 262.15 | 244.14 | 245.12 | 131.58 | 2 |
| 17 | | | | | R | 175.12 | 157.11 | 158.09 | 88.06 | 1 |

**B** Species: *M. tuberculosis*; Peptide: TQIDQVESTA**G**SLQ**G**QWR
m/z:1002.4928; z = 2; RT: 58.67; -logP = 90.83, ppm = 0.0

Species: *M. kansasii*; Peptide: TQIDQVESTA**A**SLQ**A**QWR
m/z:1017.0172; z = 2; RT: 108.27; -logP = 91.68, ppm = 7.6

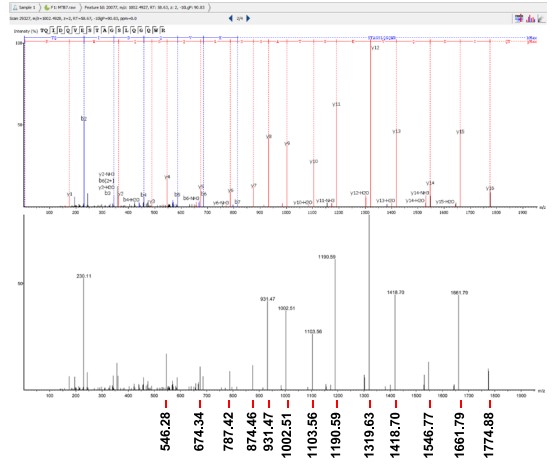
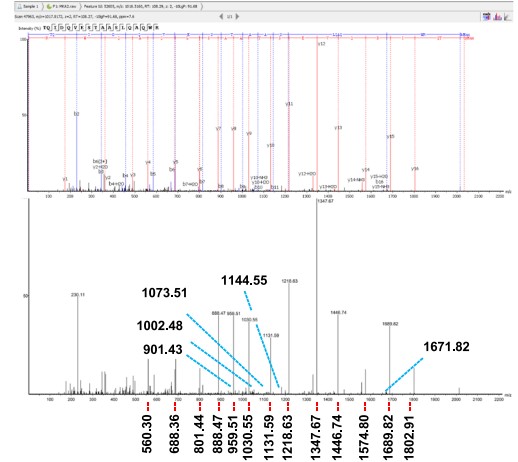

| # | b | b-H2O | b-NH3 | b (2+) | Seq | y | y-H2O | y-NH3 | y (2+) | # |
|---|---|-------|-------|--------|-----|---|-------|-------|--------|---|
| 1 | 102.06 | 84.04 | 85.03 | 51.53 | T | | | | | 18 |
| 2 | 230.11 | 212.10 | 213.09 | 115.56 | Q | 1902.93 | 1884.92 | 1885.90 | 951.97 | 17 |
| 3 | 343.20 | 325.19 | 326.17 | 172.10 | I | 1774.88 | 1756.86 | 1757.84 | 887.94 | 16 |
| 4 | 458.23 | 440.21 | 441.20 | 229.61 | D | 1661.79 | 1643.76 | 1644.76 | 831.39 | 15 |
| 5 | 586.28 | 568.27 | 569.26 | 293.64 | Q | 1546.77 | 1528.75 | 1529.72 | 773.88 | 14 |
| 6 | 685.35 | 667.34 | 668.33 | 343.16 | V | 1418.70 | 1400.70 | 1401.68 | 709.85 | 13 |
| 7 | 814.39 | 796.38 | 797.37 | 407.70 | E | 1319.63 | 1301.63 | 1302.61 | 660.32 | 12 |
| 8 | 901.43 | 883.42 | 884.40 | 451.21 | S | 1190.59 | 1172.59 | 1173.58 | 595.80 | 11 |
| 9 | 1002.47 | 984.46 | 985.45 | 501.74 | T | 1103.56 | 1085.56 | 1086.55 | 552.28 | 10 |
| 10 | 1073.51 | 1055.50 | 1056.48 | 537.26 | A | 1002.51 | 984.49 | 985.49 | 501.76 | 9 |
| 11 | 1130.53 | 1112.52 | 1113.51 | 565.77 | G | 931.47 | 913.46 | 914.45 | 466.24 | 8 |
| 12 | 1217.56 | 1199.55 | 1200.54 | 609.28 | S | 874.46 | 856.44 | 857.43 | 437.73 | 7 |
| 13 | 1330.65 | 1312.64 | 1313.62 | 665.82 | L | 787.42 | 769.41 | 770.38 | 394.21 | 6 |
| 14 | 1458.71 | 1440.70 | 1441.68 | 729.85 | Q | 674.34 | 656.31 | 657.32 | 337.67 | 5 |
| 15 | 1515.73 | 1497.72 | 1498.70 | 758.36 | G | 546.28 | 528.27 | 529.25 | 273.64 | 4 |
| 16 | 1643.79 | 1625.78 | 1626.76 | 822.39 | Q | 489.26 | 471.26 | 472.23 | 245.13 | 3 |
| 17 | 1829.87 | 1811.86 | 1812.84 | 915.43 | W | 361.20 | 343.20 | 344.17 | 181.10 | 2 |
| 18 | | | | | R | 175.12 | 157.11 | 158.09 | 88.06 | 1 |

| # | b | b-H2O | b-NH3 | b (2+) | Seq | y | y-H2O | y-NH3 | y (2+) | # |
|---|---|-------|-------|--------|-----|---|-------|-------|--------|---|
| 1 | 102.06 | 84.04 | 85.03 | 51.53 | T | | | | | 18 |
| 2 | 230.11 | 212.10 | 213.09 | 115.56 | Q | 1930.96 | 1912.95 | 1913.93 | 965.98 | 17 |
| 3 | 343.20 | 325.19 | 326.17 | 172.10 | I | 1802.91 | 1784.89 | 1785.89 | 901.95 | 16 |
| 4 | 458.23 | 440.22 | 441.20 | 229.61 | D | 1689.82 | 1671.82 | 1672.80 | 845.41 | 15 |
| 5 | 586.28 | 568.27 | 569.26 | 293.64 | Q | 1574.80 | 1556.78 | 1557.78 | 787.90 | 14 |
| 6 | 685.35 | 667.34 | 668.33 | 343.16 | V | 1446.74 | 1428.72 | 1429.72 | 723.87 | 13 |
| 7 | 814.40 | 796.38 | 797.38 | 407.70 | E | 1347.67 | 1329.66 | 1330.64 | 674.33 | 12 |
| 8 | 901.43 | 883.42 | 884.40 | 451.23 | S | 1218.63 | 1200.61 | 1201.61 | 609.81 | 11 |
| 9 | 1002.48 | 984.47 | 985.45 | 501.74 | T | 1131.59 | 1113.58 | 1114.58 | 566.30 | 10 |
| 10 | 1073.51 | 1055.50 | 1056.50 | 537.26 | A | 1030.55 | 1012.52 | 1013.53 | 515.77 | 9 |
| 11 | 1144.55 | 1126.54 | 1127.53 | 572.77 | A | 959.51 | 941.50 | 942.49 | 480.25 | 8 |
| 12 | 1231.58 | 1213.57 | 1214.55 | 616.29 | S | 888.47 | 870.46 | 871.45 | 444.73 | 7 |
| 13 | 1344.66 | 1326.65 | 1327.64 | 672.83 | L | 801.44 | 783.43 | 784.41 | 401.22 | 6 |
| 14 | 1472.72 | 1454.71 | 1455.70 | 736.86 | Q | 688.36 | 670.34 | 671.33 | 344.68 | 5 |
| 15 | 1543.76 | 1525.75 | 1526.73 | 772.38 | A | 560.30 | 542.28 | 543.27 | 280.65 | 4 |
| 16 | 1671.82 | 1653.81 | 1654.80 | 836.41 | Q | 489.26 | 471.26 | 472.23 | 245.13 | 3 |
| 17 | 1857.90 | 1839.89 | 1840.87 | 929.45 | W | 361.20 | 343.20 | 344.17 | 181.10 | 2 |
| 18 | | | | | R | 175.12 | 157.11 | 158.09 | 88.06 | 1 |

◀ **Figure EV2. Shared sequences between *M. tuberculosis* and *M. kansasii*.**

(A) The ESAT-6 peptides 'WDATATELNNALQNLAR' (left, *M. tuberculosis*) and 'WDATAQELNSALQNLSR' (right, *M. kansasii*) differ at three amino acids. (B) The CFP-10 peptides 'TQIDQVESTAGSLQGQWR' (left, *M. tuberculosis*) and 'TQIDQVESTAASLQAQWR' (righ, *M. kansasii*) differ at two amino acids. For each peptide, the top plot shows annotated y and b ions from PEAKS Studio, the middle plot displays the original MS2 spectrum, and the bottom table highlights distinct amino acids and corresponding ion transitions with green and blue frames.

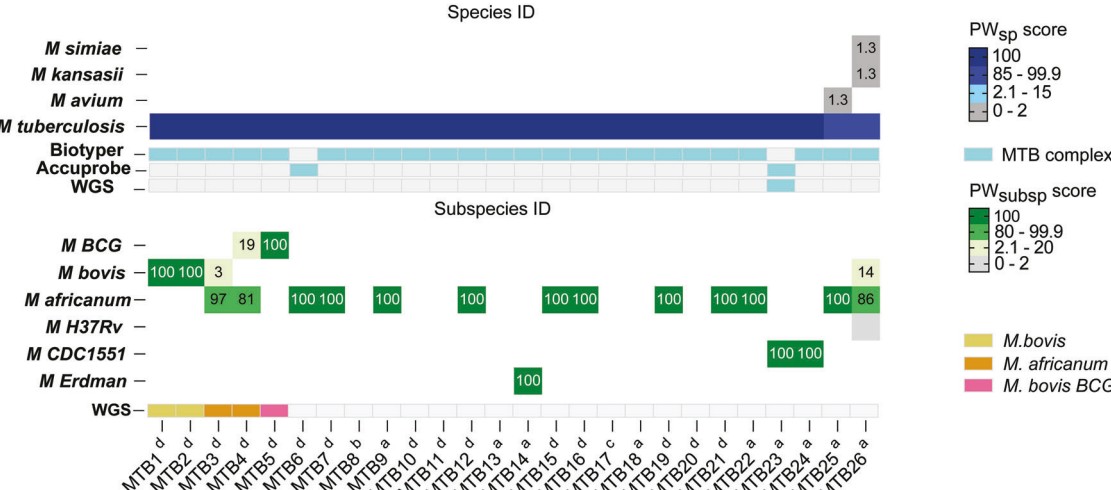

**Figure EV3.** ***M. tuberculosis* species and subspecies output from the pipeline.**

MTB species and subspecies classification using the pipeline. Samples were classified as *M. tuberculosis* or *M. tuberculosis complex* species. Nine cases, marked with an 'a,' were identified as *M. tuberculosis* based on peptides uniquely matching this species. Cases marked as 'b,' 'c,' and 'd' were classified as *M. tuberculosis* complex because the peptides identified as MTB also matched *M. canettii*, *M. orygis*, or both *M. canettii* and *M. orygis*, respectively. Subspecies identification and PWsubsp scores for each sample were determined using PEP-TORCH.

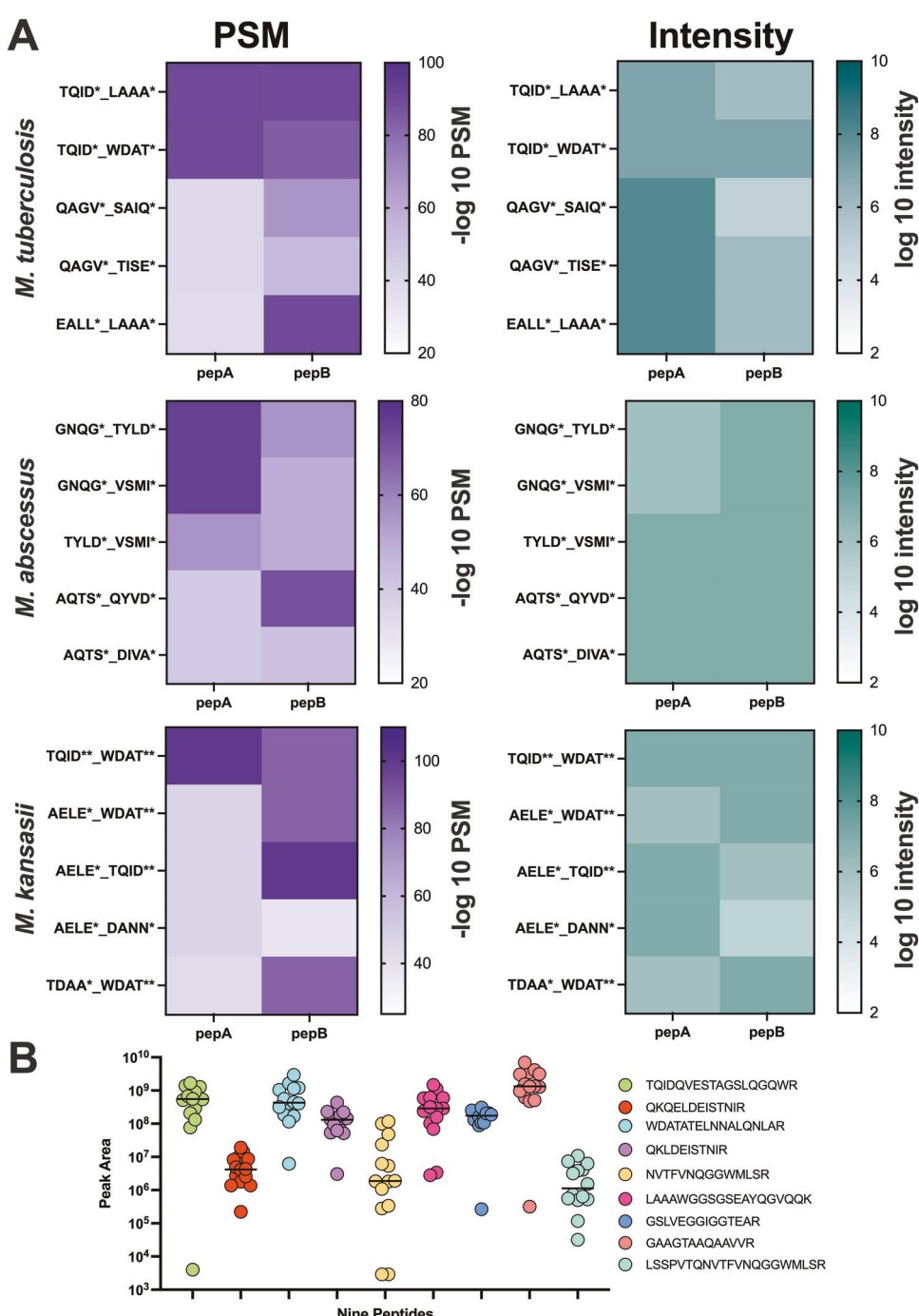

**Figure EV4. Peptide-spectral match (PSM) and peak area of individual peptides in *M. tuberculosis*, *M. abscessus* and *M. kansasii* and parallel reaction monitoring (PRM) validation of nine *M. tuberculosis* target peptides.**

(A) The median -log10 values of PSM and intensities were computed for each peptide (designated as PepA and PepB when combined) across all samples within respective species groups. Subsequently, the top five peptide combinations were identified within each species group, prioritizing those exhibiting the highest -log10 PSM and log10 intensity values. From these top combinations, three peptides were selected for further targeted analysis using PRM within each species group. The color transition from light to dark in the heat map represents the −log10 PSM values in ascending order. (B) All nine *M. tuberculosis*-specific peptides with high PSM scores (−log10 $P > 55$), that were validated by PRM in randomly chosen 14 samples. The blank line represents the mean peak area of the peptides which correspond to $10^6$–$10^8$ in average.

## A Sequence alignment of DUF5078 domain - containing protein of *M. avium* and *M. intracellulare*
## CLUSTAL O(1.2.4) multiple sequence alignment

```
tr|A0A049DNT2|A0A049DNT2_MYCAV    MSRLSRGLRAGAAFVALGVTAAIFPSTAVADSTEDFPIPRRMINTTCDAEQILAATRDTS
tr|A0A1Y0TER8|A0A1Y0TER8_MYCIT    MSRLSTGLRAGAV FLALGVTAAIFPSTAVADSTEDFPIPRRMINTTCDAEQILAATRDTS

tr|A0A049DNT2|A0A049DNT2_MYCAV    PVYYQRYMIDFNNHPNVQQATIDK AHWFYALSPQDRʀNYSENFYAPQADPLWEAWPNHMK
tr|A0A1Y0TER8|A0A1Y0TER8_MYCIT    PVYYQRYMIDFNNHPNVNQAAIDK AHWFYALSPADRʀNYSENFYAPQADPLWLAWPNHMK

tr|A0A049DNT2|A0A049DNT2_MYCAV    IFWNNKGVVAKATDICNQYPPGDMSVWNWS
tr|A0A1Y0TER8|A0A1Y0TER8_MYCIT    IFWNNKGVVAKATDICNTYPPGDMSVWNWS
```

## B Peptides mapping of *M. abscessus*

Hemophore related protein OS = *M. abscessus*
tr|A5A9S9|A5A9S9_9MYCO

MNKLSLTKTIAAVGGITMALSAGAGLASADPVTDEMVNSTCTYE
QANAALHAENPMAAEYFDASPPNQQFMREFLSSPKDKR
VSMINQVKGNQGIEYVIPVFQQMVRSCHKᵧ

Haemophore haem - binding domain - containing protein OS = *M. abscessus*
tr|R4UQX4|R4UQX4_9MYCO

MKFTSAVLSGVVGAGAVASALAFAGAADAAPSKCTAAEFAR
THSTVSSQVASYLDKNPTINDGITNAAKGAPEGQRREAIK TYLDGQPAAK
AELEKIRQPLTSLKNSCGADTDDAAPAAPAGLMGQAPAAEQPAVENAPAEQPQQPWNP
FAPQQPAPETATANTPQAAPNVAAVVDQQDV

**Figure EV5. Homology analysis in targeted peptides of *M. avium*, *M. intracellulare* and *M. abscessus*.**

(A) Sequence alignment of domain-containing protein in between *M. avium* and *M. intracellulare*. Highlighted (in red and green) are the two peptide sequence regions that show single amino acid change (highlighted in blue). (B) Three-peptide mapping to two proteins (highlighted in yellow and green) of *M. abscessus*. The red line indicates the mapped peptide sequences.

