## [Peer Review File · EMBO Molecular Medicine]

Precise Mycobacterial Species and Subspecies Identification Using the PEP-TORCH Peptidome Algorithm

Duran Bao, Sudipa Maity, Lingpeng Zhan, Seungyeon Seo, Qingbo Shu, Christopher Lyon, Bo Ning, Adrian Zelazny, Tony Hu, and Jia Fan

Corresponding author: Jia Fan (jfan5@tulane.edu)

Review Timeline:

Submission Date:	21st Aug 24
Editorial Decision:	25th Sep 24
Revision Received:	22nd Jan 25
Editorial Decision:	4th Feb 25
Revision Received:	12th Feb 25
Accepted:	19th Feb 25

Editor: Zeljko Durdevic

Transaction Report:

25th Sep 2024

Dear Dr. Fan,

Thank you for the submission of your manuscript to EMBO Molecular Medicine. We have now received feedback from the three reviewers who agreed to evaluate your manuscript. As you will see from the reports pasted below, all three referees recognize potential interest of the study but also raise important concerns that should be addressed in a major revision. If you would like to discuss further the points raised by the referees, I am available to do so via email or video. Let me know if you are interested in this option.

We would welcome the submission of a revised version within three months for further consideration. Please let us know if you require longer to complete the revision.

I look forward to receiving your revised manuscript.

Yours sincerely,

Zeljko Durdevic

We require:

- 1) A .docx formatted version of the manuscript text (including legends for main figures, EV figures and tables). Please make sure that the changes are highlighted to be clearly visible.
- 2) Individual production quality figure files as .eps, .tif, .jpg (one file per figure). For guidance, download the 'Figure Guide PDF': (<https://www.embopress.org/page/journal/17574684/authorguide#figureformat>).
- 3) A .docx formatted letter INCLUDING the reviewers' reports and your detailed point-by-point responses to their comments. As part of the EMBO Press transparent editorial process, the point-by-point response is part of the Review Process File (RPF), which will be published alongside your paper.
- 4) A complete author checklist, which you can download from our author guidelines (<https://www.embopress.org/page/journal/17574684/authorguide#submissionofrevisions>). Please insert information in the checklist that is also reflected in the manuscript. The completed author checklist will also be part of the RPF.
- 5) Please note that all corresponding authors are required to supply an ORCID ID for their name upon submission of a revised manuscript.
- 6) It is mandatory to include a 'Data Availability' section after the Materials and Methods. Before submitting your revision, primary

datasets produced in this study need to be deposited in an appropriate public database, and the accession numbers and database listed under 'Data Availability'. Please remember to provide a reviewer password if the datasets are not yet public (see <https://www.embopress.org/page/journal/17574684/authorguide#dataavailability>).

12) Author contributions: You will be asked to provide CRediT (Contributor Role Taxonomy) terms in the submission system. These replace a narrative author contribution section in the manuscript.

13) A Conflict of Interest statement should be provided in the main text.

14) Every published paper now includes a 'Synopsis' to further enhance discoverability. Synopses are displayed on the journal webpage and are freely accessible to all readers. They include a short stand first (maximum of 300 characters, including space) as well as 2-5 one-sentences bullet points that summarizes the paper. Please write the bullet points to summarize the key NEW findings. They should be designed to be complementary to the abstract - i.e. not repeat the same text. We encourage inclusion

of key acronyms and quantitative information (maximum of 30 words / bullet point). Please use the passive voice. Please attach these in a separate file or send them by email, we will incorporate them accordingly.

15) Include a Reagents and Tools Table as part of the Methods section, which can be downloaded from our author guidelines (<https://www.embopress.org/page/journal/17574684/authorguide#structuredmethods>)

**** Reviewer's comments ****

Referee #1 (Comments on Novelty/Model System for Author):

Technical quality could be high if more details were given in Methods regarding database size, strain coverage, and how their scoring is calculated and false-positives were observed or possible.

Referee #1 (Remarks for Author):

The authors Bao and colleagues had developed a tool able to perform taxonomic classification of mycobacterial culture filtrates using LC-MS/MS data. While I can foresee great relevance for this work, there are major concerns that I would like to bring to the attention of the journal and the authors before publication.

- Regarding the peptide identification: the authors claimed they used PEAKS de novo approach, but then state that a database search was performed using mycobacterial proteins. This is conflicting information, because the DB SEARCH option in PEAKS does probabilistic-based identification, not de novo as the authors says. PEAKS even generate two independent protein/peptide lists, one for option DB SEARCH and another for option DE NOVO. Considering authors are filtering their identifications based on the $-10\log P$ score, they used the probabilistic-based method, not the de novo, which has different type of scoring (the amino acid length coverage percentage, ALC%). This should be further clarified and fixed on the text. It is considered good practice in proteomics publications to provide number of proteins present in the protein sequence database and month/year it was downloaded (which counts as database version). And, to provide all MS raw data available, so readers can truly replicate the authors observations. So, I strongly recommend authors to make their data fully available using repositories such as ProteomeXchange / PRIDE.

- It is also not clear the size of the database used for taxonomical identification and for PEP-TORCH. How many strains were considered? I ask all this because authors use Unipept, which is based on Uniprot annotation, but the number of proteomes present in Uniprot is quite limited compared to number of strains in Genbank. But it is unclear in the methodology what was the strain coverage used when comparing towards Mycobacteriaceae strains with complete genomes sequenced in Genbank, for example (908 strains, 680 of those with annotated proteins). Your database contained proteins only from strains with complete, revised genomes, or it also contained proteins sequences from strains with incomplete status such as drafts, contigs, scaffolds? This is important, a quick blastp on the sequences shown in Figure EV1 for example, shows that these peptides are shared across different species: WDATATELNNALQNLAR is observed in Mtb, in Mycobacterium shinjukuense, and even on E. coli using aNCBI database. I went a step further, I downloaded the annotated proteins from those 680 strains and created a tryptic peptide library for all Mycobacteriaceae: the peptide WDATATELNNALQNLAR is seen in 276 Mtb strains, but also in M. canettii and M. orygis, in addition to M. shinjukuense listed above. Same trend is observed for the peptides used in PRM, many are identified in multiple species with a simple blastp search, I am honestly struggling to understand how it can be species-specific in this analysis. I wonder if database used had a better strain coverage, pep-torch performance would be the same, or even if peptides classified as "unique" for taxon determination would even be the same.

- It is easy to understand what a unique peptide is, but the concept of "unique pairs" or "3-peptide combinations" in text (paragraph line 128) and Figure 3B should be made clear to readers.

- The PWsp and the PWsubsp scoring calculation is explained very superficially on Results. Authors should provide more clarification on this and adding further discussion for example if there is a chance of false positives.

- The validation steps with the clinical samples and the targeted analysis seems quite good, but I think that section would be even more convincing if the authors had properly demonstrated how their "unique combinations" were selected/chosen and if database size was also clearer to the reader.

- I have some issues with some of the figures. For example:

Figures 2A and 2B: why not showing the gel so we can see the differences in depletion/enrichment? I personally think a graph to show band area % is not a good choice to visualize this: any number can be typed to generate a graph we want, but showing the real gel is undisputable evidence of enrichment/depletion.

Figure EV1, the spectra annotation is very confusing. For each peak in a spectrum, I would like to see its m/z in addition to its y/b series number, so I can infer the sequence myself and check if the identification is valid. For example, the spectra for the 1st pair shown, there is a huge peak at the lower side of the assigned y1 peak. If the m/z of that peak is 147, it indicates a K y1 and not the assigned R y1. Both peptides, even though are 3 amino acids different from each other, are not isomers only thanks to the

Ala-Ser variation on y2, because the Thr-Gln variation (-27 Da) and the Asn-Ser variation (+27 Da) null each other. Then I look at the low m/z area (which is normally noisy) and I see that the annotated y series was done to low intensity peaks and some main peaks around are not assigned, and the y2 is also assigned on a complicated area of the spectrum, it really makes me wish that I could see all m/z values of this spectra. And why the m/z of the parent ion is not given as well? Anyway, I strongly recommend that these annotated spectra should be re-done to be vastly more informative.

Also in Figure EV1: Peptide TQIDQVESTAGSLQGQWR is written as from *M. kansasii*, but that is the *Mtb* peptide accordingly to blastp. Likewise, TQIDQVESTAASLQGAWR reported as a *Mtb* protein is from *M. kansasii*.

In Figure EV2, authors claim their tool correctly identifies *M. bovis* and *M. africanum*, compared to MALDI, WGS etc which classifies the sample as *Mtb*. But isn't *bovis* and *africanum* currently classified as variants of *Mtb*? Could it be that the other approaches are identifying the samples as *Mtb* because their taxon is in line with that?

Minor concerns

- I think writing structure could be improved in some minor points, for example:

Line 47 "...mycobacterial growth indicator (MGIT) tube..." abbreviation should be placed after "tube" (after all, the T in the abbreviation stands for "tube").

Line 47 "The culture on being indicated positive for mycobacteria by the MGIT instrument is further sub-cultured..." should be simplified to "MGIT-positive cultures are further sub-cultured..."

Line 93 "Peptide Taxonomy/Organism CHecking (PEP-TORCH)", it should be written PEPTide Taxonomy/Organism CHecking (PEP-TORCH) to agree with style used in CHecking. Or just use Peptide Taxonomy/Organism CHecking (PEP-TORCH).

Lines 97-100, this last sentence really look like out of place to me personally and they resemble "methods" text. Was the authors intention to have that text by the end of Introduction?

Figure 2 Legends: Items 2C and 2D seems to be inverted (image shows # proteins then # peptides, but text says # peptides then # proteins).

Figure 3 legend, particularly item A and B are confusing and contain errors.

Referee #2 (Comments on Novelty/Model System for Author):

Ad 1) The manuscript demonstrates a rigorous approach to technical methods, especially in developing the novel PEP-TORCH algorithm. The experimental setup, mainly using LC-MS/MS, is well-detailed, and the statistical analyses are thorough. The pipeline's reproducibility, as highlighted by its performance on replicates and co-infection samples, adds to its robustness. The algorithm's validation against clinical standards (MALDI-TOF, WGS) supports the high technical quality.

Ad 2) Introducing the PEP-TORCH algorithm represents a novel approach to mycobacterial identification, particularly its ability to identify species and subspecies in parallel and detect co-infections. This is a significant advancement over conventional methods, which typically rely on sequential isolation of clinical isolates for identification. The potential to streamline the diagnostic process in a clinical setting is innovative and makes this study stand out.

Ad 3) The PEP-TORCH method has precise clinical applications, especially in improving diagnostic speed and precision for mycobacterial infections - however, the presented sample size and scope limit the immediate impact. While the method could have considerable influence in resource-limited settings, its broader adoption and utility would depend on further validation, particularly in larger cohorts and different healthcare environments. The detection of co-infections is clinically relevant, but the study's preliminary nature still limits the immediate medical impact.

Ad 4) The model system based on clinical samples of patients with suspected mycobacterial infections is appropriate for the study's aims. The samples cover a range of mycobacterial species and subspecies, and the validation against standard clinical assays provides a solid foundation. However, a more significant, diverse sample set could strengthen the conclusions. There are no ethical concerns related to the model system, as it involves anonymized clinical samples and standard microbial analysis.

Suggestions for Improvement:

- Increasing the sample size and testing the method across different healthcare facilities would further validate the model's robustness.

- The inclusion of more diverse NTM species or mixed infections could highlight the broader applicability of PEP-TORCH.

Referee #2 (Remarks for Author):

Summary: The manuscript presents a novel mycobacterial identification method, Peptide Taxonomy/Organism CHecking (PEP-TORCH). It employs a peptidome-based algorithm to accurately detect mycobacterial species and subspecies using LC-MS/MS data. The paper claims to improve diagnostic speed and precision by bypassing sub-culturing steps and identifying species and subspecies simultaneously. The study also highlights the potential of PEP-TORCH to detect co-infections. It proposes that this method could replace approaches like pure isolate culture, sequential MALDI-TOF, PCR, and targeted sequencing.

Overall Impression: The manuscript addresses a relevant problem in clinical microbiology: accurate and rapid identification of

mycobacterial species and subspecies. The presented method seems promising and offers clear improvements over existing workflows by eliminating some time-consuming steps associated with traditional culture-based identification techniques. The validation of clinical samples and comparisons with established methods like MALDI-TOF and WGS lend credibility to the technique. However, certain areas of the manuscript would benefit from further clarification and discussion. It relies on a method unavailable in many mycobacterial laboratories, which may impact applicability.

Areas of improvement:

- 1) Material and Methods: While the PEP-TORCH algorithm is described in the text, the exact mechanisms of the scoring system (PW_{sp} and PW_{subsp}) could be explained in more detail, particularly for readers unfamiliar with computational biology.
 - 2) Discussion: The manuscript briefly touches upon the clinical implementation of PEP-TORCH, but this section could be expanded. For instance, a discussion on the approach's scalability, cost, and any additional training or equipment required in clinical labs would be useful for readers from clinical settings.
 - 3) While the figures are helpful, adding a summary table comparing PEP-TORCH's performance metrics with other methods (e.g., time to result, accuracy, cost) would enhance readability.
 - 4) Figure 4 "C M. Kansasii" should say "M. kansasii".
 - 5) There are a few typographical errors that should be corrected (e.g., "The PEP-TORCH algorithm enhances mycobacterial classification by accuracy identifying species..." should read "accurately identifying species").
- Conclusion: This manuscript provides a valuable contribution to the field of microbiological diagnostics and could significantly impact the rapid identification of mycobacterial infections.

Referee #3 (Comments on Novelty/Model System for Author):

Dr Fan and colleagues develop a novel LC-MS/Algorithm capable of identifying NTM/Mtb species and sub-species. This novel technology platform is very important for clinical applications of diagnosis of NTM/Mtb. The accuracy of proof-of-concept application of such system in medium-size participants recruited from Florida are pretty impressive. Manuscript is well, clearly written.

Referee #3 (Remarks for Author):

I only have couple of minor comments for authors:

1. They focus on several key NTM species to develop and validate the system. However, can they discuss how this system may be potentially used to identify more NTM species in clinical?
2. Can they discuss more about the power of this system for differential diagnosis between NTM and BCG vaccine strain since most of people in developing countries received BCG vaccination?
3. Can they include the information for participants in Table 1 to more clearly show whether they received BCG vaccine?
4. Can they discuss more about how this system may potentially avoid the interference from other bacteria (particularly anaerobic bacteria) colonized in lungs or other extra-pulmonary organs such as intestine because anaerobic bacteria from lungs or intestine may not be easily grown by protocol shown in Figure 1. Or more generally speaking, can they discuss how this system may avoid the interference from lung/gut microbiota?

We sincerely thank all reviewers for their helpful critiques and believe that our revised manuscript has significantly improved by incorporating the suggested changes. We have addressed all the critiques in the point-by-point responses below and cited the changes that have been incorporated in the revised manuscript by “*italicized*” the text. We have also highlighted the corresponding texts in yellow in the manuscript. There are three “Response Figures,” which are only shown in this response letter. All other figures and Tables are incorporated in the revised manuscripts.

***** **Reviewer's comments** *****

Referee #1 (Comments on Novelty/Model System for Author):

Technical quality could be high if more details were given in Methods regarding database size, strain coverage, and how their scoring is calculated and false-positives were observed or possible.

Response: Thank you for your suggestions regarding database size, strain coverage, and score calculation and false positive. We have provided additional details in the following description and incorporate them into the revised manuscript.

Referee #1 (Remarks for Author):

The authors Bao and colleagues had developed a tool able to perform taxonomic classification of mycobacterial culture filtrates using LC-MS/MS data. While I can foresee great relevance for this work, there are major concerns that I would like to bring to the attention of the journal and the authors before publication.

Q1: Regarding the peptide identification: the authors claimed they used PEAKS de novo approach, but then state that a database search was performed using mycobacterial proteins. This is conflicting information, because the DB SEARCH option in PEAKS does probabilistic-based identification, not de novo as the authors says. PEAKS even generate two independent protein/peptide lists, one for option DB SEARCH and another for option DE NOVO. Considering authors are filtering their identifications based on the $-10\log P$ score, they used the probabilistic-based method, not the de novo, which has different type of scoring (the amino acid length coverage percentage, ALC%). This should be further clarified and fixed on the text. It is considered good practice in proteomics publications to provide number of proteins present in the protein sequence database and month/year it was downloaded (which counts as database version). And, to provide all MS raw data available, so readers can truly replicate the authors observations. So, I strongly recommend authors to make their data fully available using repositories such as ProteomeXchange / PRIDE.

Response Q1: Thank you for your comments. Yes, we used PEAKs DB search option for our data analysis. We have now revised the description in Methods (**Page 14, Paragraph 4**) to indicate that the peptides were identified through the “DB SEARCH” in PEAKS Studio as follows:

“Raw MS datasets were analyzed using the PEAKS DB search in PEAKS Studio 10.5 software (Bioinformatics Solutions Inc., Waterloo, Canada). Peptide sequences were searched against a protein database targeting taxonomy ID:1762 (Mycobacteriaceae), which included 2,084,438 entries from 7,108 descendant taxonomic groups. This database was downloaded from UniProtKB on August 16, 2022.”

We have deposited the raw mass spectrometry data to the ProteomeXchange Consortium via the PRIDE partner repository under the project accession number **PXD059923**. This

information has been included in the "**Data Availability**" section on **Page 16** of the revised manuscript. The data can be accessed using the following credentials:

- **Username:** reviewer_pxd059923@ebi.ac.uk
- **Password:** AvGAqUXji6dV

Q2: It is also not clear the size of the database used for taxonomical identification and for PEP-TORCH. How many strains were considered? I ask all this because authors use Unipept, which is based on Uniprot annotation, but the number of proteomes present in Uniprot is quite limited compared to number of strains in Genbank. But it is unclear in the methodology what was the strain coverage used when comparing towards Mycobacteriaceae strains with complete genomes sequenced in Genbank, for example (908 strains, 680 of those with annotated proteins). Your database contained proteins only from strains with complete, revised genomes, or it also contained proteins sequences from strains with incomplete status such as drafts, contigs, scaffolds? This is important, a quick blastp on the sequences shown in Figure EV1 for example, shows that these peptides are shared across different species: WDATATELNNALQNLAR is observed in Mtb, in Mycobacterium shinjukuense, and even on E. coli using aNCBI database. I went a step further, I downloaded the annotated proteins from those 680 strains and created a tryptic peptide library for all Mycobacteriaceae: the peptide WDATATELNNALQNLAR is seen in 276 Mtb strains, but also in M. canettii and M. orygis, in addition to M. shinjukuense listed above. Same trend is observed for the peptides used in PRM, many are identified in multiple species with a simple blastp search, I am honestly struggling to understand how it can be species-specific in this analysis. I wonder if database used had a better strain coverage, pep-torch performance would be the same, or even if peptides classified as "unique" for taxon determination would even be the same.

Response Q2: We appreciate the reviewer's detailed questions and feedback.

Database-related question: The database used for peptide identification was downloaded from UniProtKB and targeted taxonomy ID:1762 (Mycobacteriaceae). It contained 2,084,438 entries from 7,108 descendant taxonomic groups, providing broad coverage of various species and strains. The exact number of strains in the database is difficult to determine, as each of the 7,108 taxonomic groups includes a varying number of strains. For instance, Taxon ID 1768, one of these groups, consists of 12 strains, as shown in Response Figure 1. According to UniProt, "More than 95% of the protein sequences provided by UniProtKB are derived from the translation of coding sequences (CDS) submitted to the public nucleic acid databases, the EMBL-Bank/GenBank/DDBJ databases." Based on this, we understand that the database includes protein sequences derived from both complete and incomplete genomes.

Taxonomy - *Mycobacterium kansasii* (species)

Download	View proteins	View proteomes	
Mnemonic name	MYCKA	Rank	species
Taxon ID	1768	Lineage	cellular organisms > Bacteria (eubacteria) > Terrabacteria group > Actinomycetota > Actinomycetes (high G+C Gram-positive bacteria) > Mycobacteriales > Mycobacteriaceae > Mycobacterium
Scientific name	Mycobacterium kansasii	Strains	85-961 91-627 91-IT-197 ATCC 12478 / DSM 44162 / CIP 104589 / JCM 6379 / NCTC 13024 / TMC 1204 / P-16 (ATCC 12478, DSM 44162, DSM 44162T, P-16) DSM 43224 / SN 502 (DSM 43224) ATCC 14471 / NCTC 10268 / P-22 (NCTC 10268) TMH31 Fewer strains
Parent	Mycobacterium	Links	www.ncbi.nlm.nih.gov
Children	Mycobacterium kansasii ATCC 12478 Mycobacterium kansasii 662 Mycobacterium kansasii 732 Mycobacterium kansasii 824 Mycobacterium kansasii Z61 Browse all direct children (5) Browse all descendants		
Other names	Mycobacterium kansaii Mycobacterium kansasense ATCC 12478 CIP 104589 DSM 44162 4 more names		

Response Figure 1. A screenshot from UniProt shows the strains included in this 1768 taxon group.

Peptide-Species Matching and Database Results for “WDATATELNNALQNLAR”: Species identification in our analysis relies on PEP-TORCH, which detects either individual or a combination of peptides for species-specific identification. In the 26 TB cases analyzed in this study, nine were identified as *M. tuberculosis* (*Mtb*) using species-specific peptides. The remaining 17 cases were identified with peptides corresponding to three species within the MTB complex: *M. tuberculosis*, *M. canettii*, and *M. orygis*. The latter two species are rarely detected in clinical isolates (Riopel *et al*, 2024; Somoskovi *et al*, 2009). *M. orygis* is emerging as a zoonotic pathogen, while *M. canettii* is extremely rare and geographically confined. All three species belong to the MTB complex and cause tuberculosis in humans, with the same first-line drugs used for treatment. However, *M. orygis* and *M. canettii* are far less common than *M. tuberculosis*. Due to their relative clinical rarity, we initially considered all these samples as *M. tuberculosis*. We have now added annotations to **Figure EV3** with detailed information on these species and discussed this limitation in the revised manuscript (**Page 30**).

For the peptide “WDATATELNNALQNLAR,” derived from the ESAT6 protein (a known *Mtb* virulence factor), it matches *M. tuberculosis* as well as related species like *M. canettii*, *M. orygis*, and *M. shinjukuense*. Our BLAST search of the NCBI databases detected 86 matches in *Mycobacterium* (including 11 *M. shinjukuense* matches) and 2 matches in *E. coli*, which we discuss below:

- The *E. coli* matches are likely artifacts due to the presence of a 6xHis tag sequence in C-terminal of the matched protein sequence.
- *M. shinjukuense* is a rare NTM species with only 17 documented cases to date, mostly in Japan (Nakamura *et al*, 2023). However, potential *M. shinjukuense* matches were excluded by combining other identified peptides not expressed by this species in the PEP-TORCH analysis.
- Limitation: *M. canettii* and *M. orygis* belong to the *Mtb* complex, which contains several species that are closely related to *M. tuberculosis*, cause TB disease, and are treated with the same drug regimens. In some cases, we are not able to distinguish these species due to this close relationship.

Here is the revised description for MTB results in the **Page 8 Last Paragraph** and the updated **Figure EV3**.

“Finally, as shown in Fig. EV3, among the 26 TB clinical cases analyzed, nine were identified as *M. tuberculosis* based on peptides that specifically matched *M. tuberculosis*. The remaining 17 cases were classified as *M. tuberculosis* complex, as the data could not distinguish *M. tuberculosis* from *M. canettii* or *M. orygis*. Both *M. canettii* and *M. orygis* are part of the *M. tuberculosis* complex and can cause tuberculosis in humans. However, these species are rare causes of human infection within the *M. tuberculosis* complex (Riopel et al, 2024; Somoskovi et al, 2009). Detailed detection annotations are provided in Fig. EV3 and Dataset EV3J. Given the rarity of *M. orygis* and *M. canettii*, these species were considered as *M. tuberculosis* during PWsp score calculations in our analysis.”

Figure EV3. *M. tuberculosis* species and subspecies output from the pipeline. MTB species and subspecies classification using the pipeline. Samples were classified as *M. tuberculosis* or *M. tuberculosis* complex species. Nine cases, marked with an ‘a,’ were identified as *M. tuberculosis* based on peptides uniquely matching this species. Cases marked as ‘b,’ ‘c,’ and ‘d’ were classified as *M. tuberculosis* complex because the peptides identified as *M. tuberculosis* also matched *M. canettii*, *M. orygis*, or both *M. canettii* and *M. orygis*, respectively. Subspecies identification and PWsubsp scores for each sample were determined using PEP-TORCH.

PRM Target Peptides: In the original manuscript, we selected three target peptides for *M. tuberculosis* identification shown in Figure 5C. However, we targeted a broader set in the experiment, including “QKLDEISTNIR,” which is recognized by PEP-TORCH through UniPept as uniquely specific to *M. tuberculosis*. **Figure 5C** and corresponding description has been updated with this peptide in the revised manuscript (**Page 10, Paragraph 2**).

Q3: It is easy to understand what a unique peptide is, but the concept of "unique pairs" or "3-peptide combinations" in text (paragraph line 128) and Figure 3B should be made clear to readers.

Response Q3: We have revised the description of “unique pairs” and the corresponding paragraph in the manuscript to clarify this definition. We have also added descriptions of 2-peptide and 3-peptide combinations and a supplementary table as **Appendix Table S1** that

indicates examples of these combinations, and **Response Figure 2**, which employs 2- or 3-peptide combinations from **Figure 3C**.

Here is the new description in the revised manuscript on **Page 4-5**:

“The first step in the PEP-TORCH algorithm utilizes the Unipept application programming interface (API) to perform a batch analysis of all peptides identified from the CFP samples within the R Studio platform. This step generates a peptide-species matching matrix, listing peptides and their corresponding matched organisms. The next step filters the matrix to exclude peptides matching organisms that either lack annotations related to mycobacteria or are not isolated from human hosts. The Bacterial and Viral Bioinformatics Resource Center database was used to verify the human host origin of the matched mycobacteria. The remaining matches are categorized based on their ability to identify species. As shown in Fig. 3C and examples in Appendix Table S1, PEP-TORCH identifies species-events through: (1) single specific peptides, which match only one species; (2) 2-peptide combinations, where overlapping matches from two peptides identify a single species; and (3) 3-peptide combinations, where overlapping matches from three peptides identify a single species. PEP-TORCH limits multi-peptide combinations to three peptides, as further increasing the number of combinations does not significantly improve species or subspecies identification and adds a computational burden.”

Response Figure 2. This figure provides an example of specific peptides or peptide combinations used for taxon identification, based on a single *M. abscessus* case with species-level results. Dark orange indicates specific species matching, while light orange represents peptides mapped to other multiple species. Panel A shows identification using a single specific peptide, Panel B shows identification with a 2-peptide combination, and Panel C illustrates identification with a 3-peptide combination.

A complete version of the explanation is added in the revised manuscript as **Appendix Table S1** showing below:

Appendix Table S1. Detailed species and peptide matching results from Unipept, corresponding to Figure 3C.

Single Specific Peptide			
	AQTSGNPLLTSLLN		
M.abscessus	+		
2-Peptide Combination			
	AYFSTNPEAENDLR	LDTGLDKDAYQSTDFLAK	
M.abscessus	+	+	
M.chelonae	+		
M.frankinii		+	
M. sp. H001*	+		
M. sp. H002*	+		
M. sp. H054*	+		
M. sp. H072*	+		
M. sp. H092*	+		
3-Peptide Combination			
	DDVSFNETLHSYGIYAPDK	DSTDDYPVPR	GTPAQFPLGGVVPQFK
M.abscessus	+	+	+
M.Franklinii	+		+
M.immunogenum		+	+
M. saopaulense	+	+	
M. salmoniphilium	+	+	
M. sp. CBMA 271	+		
M. sp. LB1	+		+
M. sp. H001*	+	+	
M. sp. H002*	+	+	
M. sp. H054*	+	+	
M. sp. H072*	+	+	
M. sp. H092*	+	+	
M. stephanolepidis*	+		

*, Species identified through Unipept matching but with no recorded evidence of human infection according to BV-BRC.

Q4: The PWsp and the PWsubsp scoring calculation is explained very superficially on Results. Authors should provide more clarification on this and adding further discussion for example if there is a chance of false positives.

Response Q4: We acknowledge the lack of clarity regarding the PWsp and PWsubsp scoring approach and have addressed this by adding a detailed explanation in **Appendix Table S2**, which is now cited in the **Methods section (Page 16)** of the revised manuscript. This explanation outlines how species or subspecies scores are calculated using single-peptide, two-

peptide, and three-peptide identification events, normalized by the total detection events across all taxa. **Appendix Table S2** also provides four examples of PW_{sp} and PW_{subsp} calculations to illustrate the method.

Our validation sample set results show that the predominant scores have no false positives. However, for minor scores in samples with fewer than 100 PW_{sp}, additional clinical samples with matching WGS data are needed for further validation. If necessary, cut-off scores can be established in future studies with larger validation sample sets.

Appendix Table S2:

PEPTORCH Score Calculation Equation Explanation:

PEPTORCH Score Calculation Equation
We calculate the PEPTORCH Score for each taxon (e.g., species, subspecies, or strains) based on species-specific peptide identification events recognized by the PEPTORCH algorithm. The score for each taxon is: $\text{Score}(a) = \frac{(E_{a1} + E_{a2} + E_{a3})}{t} \times 100\%$ $\text{Score}(b) = \frac{(E_{b1} + E_{b2} + E_{b3})}{t} \times 100\%$ ... .. Where:  - a, b, ... : Different taxa (species, subspecies, or strains) identified by species-specific peptides. - E: An identification event triggered by single specific peptide or peptide combinations in the PEPTORCH algorithm for a given taxon. - E_{a1}: The total number of single peptide identification events for taxon a. - E_{a2}: The total number of two-peptide combination events for taxon a. - E_{a3}: The total number of three-peptide combination events for taxon a. - t: Total number of identification events across all taxa ($t = \sum E_{x1} + \sum E_{x2} + \sum E_{x3}$) Each taxon's PEPTORCH Score represents the percentage of its species peptide events relative to the total events from all taxa.
Example 1: Single infection (MAB4)
 • Taxa: M. abscessus (mab) • Input Peptides: 20 • Species Peptide Events:  ○ Single-peptide events (E_{mab1}): 3 ○ Two-peptide events (E_{mab2}): 61 ○ Three-peptide events (E_{mab3}): 460 • Total Events, t: $t = E_{mab1} + E_{mab2} + E_{mab3} = 3 + 61 + 460 = 524$ • Score: $\text{Score}(mab) = \frac{(3 + 61 + 460)}{524} \times 100\% = 100\%$ • Result: The PEP-TORCH pipeline scores at 100% for the unknown sample as M. abscessus.
Example 2: (co-infection)
 • Taxa: M. avium (mav) and M. kansasii (mkan) • Input Peptides: 47 • Species Peptide Events:

- For *M. avium*:
 1. (E_{mav1}): 1
 2. (E_{mav2}): 45
 3. (E_{mav3}): 994
- For *M. kansasii*:
 1. (E_{mkan1}): 1
 2. (E_{mkan2}): 20
 3. (E_{mkan3}): 119
- **Total Events, t:**

$$t = (E_{mab1} + E_{mav2} + E_{mav3}) + (E_{mkan1} + E_{mkan2} + E_{mkan3}) = (1 + 45 + 994) + (1 + 20 + 119) = 1180$$

- **Scores:**

$$Score(mav) = \frac{(1 + 45 + 994)}{1180} \times 100\% = 88.1\%$$

$$Score(mkan) = \frac{(1 + 20 + 119)}{1180} \times 100\% = 11.9\%$$

- **Result:** The PEP-TORCH pipeline identifies this samples as *M. avium* at 88.1% and *M. kansasii* at 11.9%.

Example 3: Subspecies Analysis (MAB3)

- **Taxa:** *M. abscessus subsp. massiliense* (submass)
- **Input Peptides:** 33
- **Subspecies Peptide Events:**
 - Single-peptide events ($E_{submass1}$): 2
 - Two-peptide events ($E_{submass2}$): 63
 - Three-peptide events ($E_{submass3}$): 961

- **Total Events, t:**

$$t = E_{submass1} + E_{submass2} + E_{submass3} = 3 + 61 + 460 = 1026$$

- **Score:**

$$Score(submass) = \frac{(2 + 63 + 961)}{1026} \times 100\% = 100\%$$

- **Result:** The PEP-TORCH pipeline scores at **100%** for the unknown sample as *M. abscessus subsp. massiliense*.

Example 4: Subspecies Analysis (MAB22)

- **Taxa:** *M. abscessus subsp. abscessus* (subab) and *M. abscessus subsp. massiliense* (submass)
- **Input Peptides:** 33
- **Subspecies Peptide Events:**

- For subab
 1. Single-peptide events (E_{subab1}): 2
 2. Two-peptide events (E_{subab2}): 139
 3. Three-peptide events (E_{subab3}): 4691
- For submass
 1. Single-peptide events ($E_{submass1}$): 1
 2. Two-peptide events ($E_{submass2}$): 59
 3. Three-peptide events ($E_{submass3}$): 1711

- **Total Events, t:**

$$t = E_{subab1} + E_{subab2} + E_{subab3} + E_{submass1} + E_{submass2} + E_{submass3} = 2 + 139 + 4691 + 1 + 59 + 1711 = 6603$$

- **Score:**

$$Score(subab) = \frac{(2 + 139 + 4691)}{6603} \times 100\% = 73.2\%$$

$$\text{Score}(\text{submass}) = \frac{(1 + 59 + 1711)}{6603} \times 100\% = 26.8\%$$

- **Result:** The PEP-TORCH pipeline identifies this samples as *M. abscessus subsp. abscessus* at 73.2% and *M. abscessus subsp. massiliense* at 26.8%.

Q5: The validation steps with the clinical samples and the targeted analysis seems quite good, but I think that section would be even more convincing if the authors had properly demonstrated how their "unique combinations" were selected/chosen and if database size was also clearer to the reader.

Response Q5: We have added more details about the database size and multiple-peptide combinations in the revised manuscript in response to **Q1–Q4**. Additionally, we have updated the figures and legends and included new supplementary figures, tables, and datasets to address these points comprehensively. Beyond peptide-species matching, we also considered other factors when selecting targeted peptides for PRM analysis (Figure 5A and B), such as detection frequency and mass spectrometry performance. Further details on the peptides analyzed in the PRM study are now provided below.

***M. tuberculosis* PRM peptides:** Based on Unipept results, the three *M. tuberculosis* PRM peptides analyzed in the original Figure 5C matched *M. tuberculosis* strains as well as *M. canettii* and *M. orygis*. However, *M. canettii* and *M. orygis* are rarely detected in clinical isolates. Genome data from BV-BRC includes 19,602 genomes of *M. tuberculosis* from human isolates, compared to just 35 genomes of *M. canettii* and 2 genomes of *M. orygis*. Although these numbers do not directly reflect epidemiological prevalence, they underscore the rarity of *M. canettii* and *M. orygis* in comparison to *M. tuberculosis*.

In our original experiment, we tested more than three *M. tuberculosis*-targeted peptides, including a single specific peptide (QKLDEISTNIR) that matched only *M. tuberculosis* strains. In the revised manuscript, this peptide has been added to **Figure 5C**. As shown in the updated **Dataset EV4B**, incorporating this peptide allows for the specific identification of *M. tuberculosis* species.

Dataset EV4B. Species matching the *M. tuberculosis* PRM peptide targets.

Species Complex	Species	TQIDQVESTAGSLQGQWR	WDATATELNALQNLAR	LAAAWGGSGSEAYQGVQK	QKLDEISTNIR
M. tuberculosis Complex	M. tuberculosis	+	+	+	+
	M. orygis	+	+	+	
	M. canetti	+	+	+	
M. simiae Complex	M. simiae		+		
M. kansasii Complex	M. attenuatum			+	
	M. gastri			+	
	M. innocens			+	
	M. ostraviense			+	
	M. persicum			+	
	M. pseudokansasii			+	
	M. kiyosense			+	
	M. riyadhense			+	

	M. simulans			+	
	M. sp. 1423905.2			+	
	M. shinjukuense		+	+	
	M. ulcerans			+	
	M. asiaticum			+	

***M. abscessus* PRM Peptides:** Based on Unipept matching, the three *M. abscessus* targeted peptides were primarily detected in *M. abscessus* (**Dataset EV4D**) but were also detected in *M. franklinii*, a member of the *M. chelonae/M. abscessus* complex, which is very rarely detected in clinical isolates with the first reported cases in 2011 (26 cases from a multicenter investigation) and an additional case reported in 2023 (Simmon *et al*, 2011; Wang *et al*, 2023). Given the rarity of *M. franklinii* cases, these three peptides remain strong indicators for *M. abscessus* detection.

These three peptides were initially selected based on their detection frequency in clinical isolates and mass spectrometry performance, but **we have since added two more peptides** that individually identify *M. abscessus* as PRM targets. These total five *M. abscessus* PRM peptides were validated with 10 additional *M. abscessus* cases, as shown in revised **Figure 5C**.

Dataset EV4D. Species matching the *M. abscessus* PRM peptide targets.

Species Complex	Species	VSMINQVK	GNQGEYVIPVFQQMVR	TYLDGQPAAK	AQTSGNPLLTSLN	LIGFDTNEAVSHGPVEVK
M. abscessus complex	M. abscessus	+	+	+	+	+
	M. franklinii	+	+	+		
	M. chelonae		+	+		
	M. saopaulense		+	+		
	M. salmoniphilium			+		
	M. sp. LB1		+	+		
	M. sp. CBMA 271			+		
	M. sp. H001*		+	+		
	M. sp. H002*		+	+		
	M. sp. H054*		+	+		
	M. sp. H072*		+	+		
	M. sp. H092*		+	+		
	M. stephanolepidis*		+	+		

*, Species identified through Unipept matching but with no recorded evidence of human infection according to BV-BRC.

***M. Kansasii* PRM Peptides:** The three *M. kansasii* peptides selected as PRM targets primarily matched *M. kansasii* isolates, and while these peptides also matched *M. gastri*, which belongs to the *M. kansasii* complex, this species is rarely associated with human disease and sometimes considered non-pathogenic (Perandones *et al*, 1991; Shahraki *et al*, 2017; Velayati *et al*, 2005). These three peptides also match *M. persicum*. Infections caused by *M. persicum* in humans are rare, with only a few confirmed cases reported, including four pulmonary infections initially identified in Iran and one case of septic arthritis in the United States (Dumais *et al*, 2023; Shahraki *et al*, 2017). Given the rarity of *M. gastri* and *M. persicum* cases, these three peptides remain strong indicators for *M. Kansasii* detection. We have discussed this limitation in the **Discussion** of the revised manuscript (**Page 13, last paragraph**).

Dataset EV4F. Species matching the *M. kansasii* PRM peptide targets.

Species Complex	Species	TQIDQVESTAASLQAQWR	AELEEISTNIR	WDATAQELNNALQNLRSR
-----------------	---------	--------------------	-------------	--------------------

M. kansasii complex	M. kansasii	+	+	+
	M. attenuatum	+	+	
	M. gastri	+	+	+
	M. innocens	+	+	
	M. ostraviense*	+	+	+
	M. persicum	+	+	+
	M. pseudokansasii	+		
	M. basiliense		+	
M. simiae complex	M. simiae	+		

*, Species identified through Unipept matching but with no recorded evidence of human infection according to BV-BRC.

PRM Peptides for *M. intracellulare* and *M. avium*: PRM peptides for *M. intracellulare* and *M. avium* were selected based on their mass spectrometry performance (**Figure 5B**). The two *M. intracellulare* peptide targets primarily matched *M. intracellulare*, but not *M. avium* (**Dataset EV4H, J**) when compared to the UniPept database entries. However, these three peptides also matched *M. paraintracellulare* and *M. timonense* isolates. Both of which belong to the MAC complex, but a recent study has suggested that *M. paraintracellulare* should be reclassified as an *M. intracellulare* subspecies (Tateishi *et al*, 2021). Both these species are rarely in clinical isolates, and *M. timonense* has not been detected and reported as a human pathogen since its first description in 2009 (Zurita *et al*, 2014). These PRM peptides are thus still likely to accurately identify *M. intracellulare*, which is a common cause of mycobacterial respiratory infection.

Dataset EV4J. Species matching the *M. intracellulare* PRM peptide targets.

Species Complex	Species	NYSEFYAPQADPLWLAWPNHMK	NVLGHILTAANADVNLALYQWWR
M. avium complex (MAC)	M. intracellulare	+	+
	M. paraintracellulare	+	+
	M. timonense	+	+
	M. sp.MOTT36Y*	+	+
	M. sp. 852002-10029 SCH5224772		+

*The species were returned from the Unipept website, but they were non-human hosted.

The three peptides selected as PRM targets for *M. avium* were primarily detected in *M. avium* isolates, including a *M. avium* complex strain (*Mycobacterium sp. MAC_011194_8550*) reported as a human isolate with no further records. We thus believe this three-peptide combination is suitable for species-specific identification of *M. avium*.

Dataset EV4H. Species matching the *M. avium* PRM peptide targets.

Species Complex	Species	HPDLHQQLQQR	AHWFYALSPQDR	AAGAGATVLNVSK
M. avium complex (MAC)	M. avium	+	+	+
	M.sp.MAC_011194_8550	+	+	+
	M. colombiense	+		
	M. marseiliense		+	
	M.sp.1165178.9	+		
	M.sp.1245801.1	+		

	M.sp.1245852.3	+		
	M.sp.1482292.6	+		
	M.sp.852002-51971_SCH5477799-a	+		
	M.sp.852002-10029_SCH5224772		+	
	M.sp.E2479	+		

I have some issues with some of the figures. For example:

Q6: Figures 2A and 2B: why not showing the gel so we can see the differences in depletion/enrichment? I personally think a graph to show band area % is not a good choice to visualize this: any number can be typed to generate a graph we want, but showing the real gel is undisputable evidence of enrichment/depletion.

Response Q6: We have added the original gel images to the revised manuscript as **Figure EV1**, as well as a summary of the relative enrichment of <60 kDa proteins in the CFP aliquots collected before and after a size-selective protein precipitation procedure, as determined by mass spectrometry protein identifications.

Figure EV1. Characterization of MGIT CFP samples before and after precipitation. A. SDS-PAGE analysis of protein size distributions in six MGIT CFP supernatant samples before (PP-) and after (PP+) precipitation with 50% acetonitrile, analyzed by gels stained with Coomassie blue. The 120 kDa, 66 kDa, 40 kDa, and 12 kDa protein markers are highlighted. B. LC-MS/MS analysis of these six CFP samples showed at least a two-fold increase in the number of low molecular weight proteins (<60 kDa) with enriched intensities following precipitation.

Q7: Figure EV1, the spectra annotation is very confusing. For each peak in a spectrum, I would like to see its m/z in addition to its y/b series number, so I can infer the sequence myself and check if the identification is valid. For example, the spectra for the 1st pair shown, there is a huge peak at the lower side of the assigned y1 peak. If the m/z of that peak is 147, it indicates a K y1 and not the assigned R y1. Both peptides, even though are 3 amino acids different from each other, are not isomers only thanks to the Ala-Ser variation on y2, because the Thr-Gln variation (-27 Da) and the Asn-Ser variation (+27 Da) null each other. Then I look at the low

m/z area (which is normally noisy) and I see that the annotated y series was done to low intensity peaks and some main peaks around are not assigned, and the y2 is also assigned on a complicated area of the spectrum, it really makes me wish that I could see all m/z values of this spectra. And why the m/z of the parent ion is not given as well? Anyway, I strongly recommend that these annotated spectra should be re-done to be vastly more informative.

Response Q7: Thank you for your detailed review and valuable suggestions. We have updated the representative MS2 spectra in **Figure EV2** and included the m/z values for the parent ions and the Ion Match Table.

Two updated figure pairs are included in the revised manuscript and have been attached to this response for your review. All annotated spectra were directly exported from PEAKS Studio. The "Ion Match Table" also displays calculated ion masses exported from PEAKS Studio. Fragment ions are highlighted in color: blue for N-terminal ions and red for C-terminal ions. Ions that differ between the two homologous sequences are highlighted in bold. Most of the y ions differ between these homologs, and were detected, while fewer b ions differ since, as noted by the reviewer, "Thr-Gln variation (-27 Da) and Asn-Ser variation (+27 Da)" nullify each other in the WDATA... peptide. Nonetheless, more than half of the distinct b ions were observed.

Here is the updated **Figure EV1** in the revised manuscript:

Figure EV2. Shared Sequences Between *M. kansasii* and *M. tuberculosis*.
 A. The ESAT-6 peptides 'WDATATELNNALQNLAR' (left, *M. tuberculosis*) and 'WDATAQELNSALQNLSR' (right, *M. kansasii*) differ at three amino acids. B. The CFP-10 peptides 'TQIDQVESTAGSLQGQWR' (left, *M. tuberculosis*) and 'TQIDQVESTAASLQAQWR' (right, *M. kansasii*) differ at two amino acids. For each peptide, the top plot shows annotated y and b ions from PEAKS Studio, the middle plot displays the original MS2 spectrum, and the bottom table highlights distinct amino acids and corresponding ion transitions with green and blue frames.

Q8: Also in Figure EV1: Peptide TQIDQVESTAGSLQGQWR is written as from *M. kansasii*, but that is the *Mtb* peptide accordingly to blastp. Likewise, TQIDQVESTAASLQGAWR reported as a *Mtb* protein is from *M. kansasii*.

Response Q8: Thank you for catching this labeling error, which we have corrected in the updated **Figure EV2**.

Q9: In Figure EV2, authors claim their tool correctly identifies *M. bovis* and *M. africanum*, compared to MALDI, WGS etc which classifies the sample as *Mtb*. But isn't *bovis* and *africanum* currently classified as variants of *Mtb*? Could it be that the other approaches are identifying the samples as *Mtb* because their taxon is in line with that?

Response Q9: Yes. MALDI-Biotyper identify the samples as *M. tuberculosis* complex, which includes *M. tuberculosis*. UniProt classifies *M. bovis* and *M. africanum* as subspecies of *M. tuberculosis*, as shown in **Response Figure 3**. Methods that provide higher taxonomic resolution can deliver valuable epidemiological insights and support treatment decisions. Thus, we highlighted our method's ability to classify all samples at the species and subspecies level.

Response Figure 3. The diagram illustrates the organization of taxonomic groups.

Minor concerns

Q10: I think writing structure could be improved in some minor points, for example:

Line 47 "...mycobacterial growth indicator (MGIT) tube..." abbreviation should be placed after "tube" (after all, the T in the abbreviation stands for "tube").

Response Q10: Thank you for pointing out this labeling error. We have corrected it in revised manuscript.

Q11: Line 47 "The culture on being indicated positive for mycobacteria by the MGIT instrument is further sub-cultured..." should be simplified to "MGIT-positive cultures are further sub-cultured...."

Response Q11: Thank you for pointing out this error. We have made the necessary adjustments in the revised manuscript.

Q12: Line 93 "Peptide Taxonomy/Organism CHecking (PEP-TORCH)", it should be written PEptide Taxonomy/Organism CHecking (PEP-TORCH) to agree with style used in CHecking. Or just use Peptide Taxonomy/Organism Checking (PEP-TORCH).

Response Q12: Thank you for your thorough review. We have updated the writing as "PEptide Taxonomy/Organism CHecking (PEP-TORCH)" in the revised version.

Q13: Lines 97-100, this last sentence really look like out of place to me personally and they resemble "methods" text. Was the authors intention to have that text by the end of Introduction?

Response Q13: Thank you for taking the time to provide the thoughtful comment, and we have revised the last paragraph of **Introduction**, in **Page 4-5**:

"To overcome the limitations of detection of mycobacteria through prolonged solid culture methods, we developed a streamlined process to analyze culture filtrate protein (CFP) samples from MGIT growth cultures using LC-MS/MS, coupled with an automated PEptide Taxonomy/Organism CHecking (PEP-TORCH) pipeline. This pipeline identifies species- and subspecies-specific mycobacterial peptide signatures. A comparison of workflow between standard clinical protocols and the PEP-TORCH method is shown in Fig. 1. The MALDI-TOF Biotyper, the current clinical standard provides species- or species-complex-level identification, however, subspecies-level identification typically requires whole-genome sequencing (WGS). In contrast, PEP-TORCH delivers taxonomy scores for identified species and subspecies with reduced turnaround times by eliminating the need for solid culture. In this study, 102 samples (Table 1) were analyzed, with 81 validated using the PEP-TORCH pipeline. Additionally, 63 samples underwent targeted proteomics guided by PEP-TORCH selection, including 42 validated by both methods and 21 exclusively through the targeted approach. The PEP-TORCH pipeline offers a comprehensive and efficient one-stop solution for species- and subspecies-level identification in NTM diagnostics."

Q14: Figure 2 Legends: Items 2C and 2D seems to be inverted (image shows # proteins then # peptides, but text says # peptides then # proteins).

Response Q14: Thanks for your pointing out the error, we have corrected those numbers in the revised manuscript.

Q15: Figure 3 legend, particularly item A and B are confusing and contain errors.

Response Q15: We have revised the entire figure 3 legend shown as below:

“Figure 3. (A) Tryptic peptides identified by mass spectrometry from MGIT filtrate samples were processed by PEP-TORCH to generate species and subspecies detection results, including peptide weightage scores. (B) PEP-TORCH processing includes filtering steps to exclude peptides not matching mycobacteria or without recorded human hosts in the Bacterial and Viral Bioinformatics Resource Center (BV-BRC). (C) The decision tree algorithm for taxon identification relies on single-species peptides, as well as combinations of two or three species-specific peptides, to determine species or subspecies. All panels were created using BioRender.com.”

Referee #2 (Comments on Novelty/Model System for Author):

Ad 1) The manuscript demonstrates a rigorous approach to technical methods, especially in developing the novel PEP-TORCH algorithm. The experimental setup, mainly using LC-MS/MS, is well-detailed, and the statistical analyses are thorough. The pipeline's reproducibility, as highlighted by its performance on replicates and co-infection samples, adds to its robustness. The algorithm's validation against clinical standards (MALDI-TOF, WGS) supports the high technical quality.

Ad 2) Introducing the PEP-TORCH algorithm represents a novel approach to mycobacterial identification, particularly its ability to identify species and subspecies in parallel and detect co-infections. This is a significant advancement over conventional methods, which typically rely on sequential isolation of clinical isolates for identification. The potential to streamline the diagnostic process in a clinical setting is innovative and makes this study stand out.

Ad 3) The PEP-TORCH method has precise clinical applications, especially in improving diagnostic speed and precision for mycobacterial infections - however, the presented sample size and scope limit the immediate impact. While the method could have considerable influence in resource-limited settings, its broader adoption and utility would depend on further validation, particularly in larger cohorts and different healthcare environments. The detection of co-infections is clinically relevant, but the study's preliminary nature still limits the immediate medical impact.

Ad 4) The model system based on clinical samples of patients with suspected mycobacterial infections is appropriate for the study's aims. The samples cover a range of mycobacterial species and subspecies, and the validation against standard clinical assays provides a solid foundation. However, a more significant, diverse sample set could strengthen the conclusions. There are no ethical concerns related to the model system, as it involves anonymized clinical samples and standard microbial analysis.

Suggestions for Improvement:

- Increasing the sample size and testing the method across different healthcare facilities would further validate the model's robustness.
- The inclusion of more diverse NTM species or mixed infections could highlight the broader applicability of PEP-TORCH.

Response: Thank you for the reviewer's comments regarding the "high technical quality" and "innovative" aspects of our work. We also appreciate the suggestion for improvement. We agree that increasing the number and species diversity of the analysis samples would be valuable. To this end, during the revision process, we obtained an additional 29 samples, including 27 MGIT culture filtrate samples from the clinical microbiology laboratory at the NIH Clinical Center and 2 isolates from The BEI Resources Repository, which were cultured in our BSL2 laboratory. Of the 29 samples, 24 are from the most common clinical species: *M. abscessus*, *M. intracellulare*, and *M. avium*, while the remaining 5 are from less prevalent species: *M. goodnae*, *M. mucogenicum*, *M. fortuitum*, *M. simiae*, and *M. floretinum*.

The results from these 29 additional samples have been incorporated into the revised manuscript, including updates to **Figures 4 and 5**, **Tables 1-3**, and the corresponding **Datasets**.

Referee #2 (Remarks for Author):

Summary: The manuscript presents a novel mycobacterial identification method, Peptide Taxonomy/Organism CHecking (PEP-TORCH). It employs a peptidome-based algorithm to

accurately detect mycobacterial species and subspecies using LC-MS/MS data. The paper claims to improve diagnostic speed and precision by bypassing sub-culturing steps and identifying species and subspecies simultaneously. The study also highlights the potential of PEP-TORCH to detect co-infections. It proposes that this method could replace approaches like pure isolate culture, sequential MALDI-TOF, PCR, and targeted sequencing.

Overall Impression: The manuscript addresses a relevant problem in clinical microbiology: accurate and rapid identification of mycobacterial species and subspecies. The presented method seems promising and offers clear improvements over existing workflows by eliminating some time-consuming steps associated with traditional culture-based identification techniques. The validation of clinical samples and comparisons with established methods like MALDI-TOF and WGS lend credibility to the technique. However, certain areas of the manuscript would benefit from further clarification and discussion. It relies on a method unavailable in many mycobacterial laboratories, which may impact applicability.

Areas of improvement:

Q1: Material and Methods: While the PEP-TORCH algorithm is described in the text, the exact mechanisms of the scoring system (PW_{sp} and PW_{subsp}) could be explained in more detail, particularly for readers unfamiliar with computational biology.

Response Q1: We have added **Appendix Table S2** to the revised manuscript to explain the score calculation approach in more detail, and provide 4 examples from our clinical cases as follows:

PEPTORCH Score Calculation Equation

We calculate the **PEPTORCH Score** for each taxon (e.g., species, subspecies, or strains) based on species-specific peptide identification events recognized by the PEPTORCH algorithm. The score for each taxon is:

$$\text{Score}(a) = \frac{(E_{a1} + E_{a2} + E_{a3})}{t} \times 100\%$$

$$\text{Score}(b) = \frac{(E_{b1} + E_{b2} + E_{b3})}{t} \times 100\%$$

... ..

Where:

- **a, b, ...** : Different taxa (species, subspecies, or strains) identified by species-specific peptides.
- **E**: An identification event triggered by single peptide or unique peptide combinations in the PEPTORCH algorithm for a given taxon.
- **E_{a1}**: The total number of single peptide identification events for taxon **a**.
- **E_{a2}**: The total number of two-peptide combination events for taxon **a**.
- **E_{a3}**: The total number of three-peptide combination events for taxon **a**.
- **t**: Total number of identification events across **all** taxa ($t = \sum E_{x1} + \sum E_{x2} + \sum E_{x3}$)

Each taxon's PEPTORCH Score represents the percentage of its species peptide events relative to the total events from all taxa.

Example 1: Single infection (MAB4):

- **Taxa:** *M. abscessus* (mab)
- **Input Peptides:** 20

- **Species Peptide Events:**
 - Single-peptide events (E_{mab1}): 3
 - Two-peptide events (E_{mab2}): 61
 - Three-peptide events (E_{mab3}): 460

- **Total Events, t:**

$$t = E_{mab1} + E_{mab2} + E_{mab3} = 3 + 61 + 460 = 524$$

- **Score:**

$$\text{Score}(mab) = \frac{(3 + 61 + 460)}{524} \times 100\% = 100\%$$

- **Result:** The PEP-TORCH pipeline scores at **100%** for the unknown sample as *M. abscessus*.

Example 2: (co-infection)

- **Taxa:** *M. avium* (mav) and *M. kansasii* (mkan)
- **Input Peptides:** 47
- **Species Peptide Events:**

- For *M. avium*:
 1. (E_{mav1}): 1
 2. (E_{mav2}): 45
 3. (E_{mav3}): 994
- For *M. kansasii*:
 1. (E_{mkan1}): 1
 2. (E_{mkan2}): 20
 3. (E_{mkan3}): 119

- **Total Events, t:**

$$t = (E_{mab1} + E_{mav2} + E_{mav3}) + (E_{mkan1} + E_{mkan2} + E_{mkan3}) = (1 + 45 + 994) + (1 + 20 + 119) = 1180$$

- **Scores:**

$$\text{Score}(mav) = \frac{(1 + 45 + 994)}{1180} \times 100\% = 88.1\%$$

$$\text{Score}(mkan) = \frac{(1 + 20 + 119)}{1180} \times 100\% = 11.9\%$$

- **Result:** The PEP-TORCH pipeline identifies this samples as *M. avium* at 88.1% and *M. kansasii* at 11.9%.

Example 3: Subspecies Analysis (MAB3)

- **Taxa:** *M. abscessus subsp. massiliense* (submass)
- **Input Peptides:** 33
- **Subspecies Peptide Events:**

- Single-peptide events ($E_{submass1}$): 2
- Two-peptide events ($E_{submass2}$): 63
- Three-peptide events ($E_{submass3}$): 961

- **Total Events, t:**

$$t = E_{submass1} + E_{submass2} + E_{submass3} = 2 + 63 + 961 = 1026$$

- **Score:**

$$\text{Score}(submass) = \frac{(2 + 63 + 961)}{1026} \times 100\% = 100\%$$

- **Result:** The PEP-TORCH pipeline scores at **100%** for the unknown sample as *M. abscessus subsp. massiliense*.

Example 4: Subspecies Analysis (MAB22)

- **Taxa:** *M. abscessus subsp. abscessus* (subab) and *M. abscessus subsp. massiliense* (submass)
- **Input Peptides:** 33

- **Subspecies Peptide Events:**
 - For subab
 1. Single-peptide events (E_{subab1}): 2
 2. Two-peptide events (E_{subab2}): 139
 3. Three-peptide events (E_{subab3}): 4691
 - For submass
 1. Single-peptide events ($E_{submass1}$): 1
 2. Two-peptide events ($E_{submass2}$): 59
 3. Three-peptide events ($E_{submass3}$): 1711
- **Total Events, t:**

$$t = E_{subab1} + E_{subab2} + E_{subab3} + E_{submass1} + E_{submass2} + E_{submass3} = 2 + 139 + 4691 + 1 + 59 + 1711 = 6603$$
- **Score:**

$$\text{Score}(subab) = \frac{(2 + 139 + 4691)}{6603} \times 100\% = 73.2\%$$

$$\text{Score}(submass) = \frac{(1 + 59 + 1711)}{6603} \times 100\% = 26.8\%$$
- **Result:** The PEP-TORCH pipeline identifies this samples as *M. abscessus subsp. abscessus* at 73.2% and *M. abscessus subsp. massiliense* at 26.8%.”

Q2: Discussion: The manuscript briefly touches upon the clinical implementation of PEP-TORCH, but this section could be expanded. For instance, a discussion on the approach's scalability, cost, and any additional training or equipment required in clinical labs would be useful for readers from clinical settings.

Response Q2: Thank you for the valuable suggestion. We have added discussion about the scalability, cost, and requirements for clinical adoption in the revised manuscript.

PEP-TORCH eliminates the need for extended culture times, offering a significant time-saving advantage over traditional solid culture-based methods like MALDI-Biotyper. In our validation sample set, the turnaround time was reduced by an average of 13 days (**Figure 4B**), enabling faster diagnosis and treatment. Additionally, PEP-TORCH is being developed into a user-friendly, one-click software that simplifies data processing and reduces the need for highly experienced technicians. This makes the assay easier to integrate into existing clinical laboratories and minimizes training requirements.

While TB remains a significant burden in resource-limited countries, NTM infections have risen notably in developed countries in recent decades (Conyers & Saunders, 2024; Dartois & Dick, 2024; Menzies *et al*, 2021). This trend makes PEP-TORCH particularly suited for centralized labs in these regions. Our targeted mass spectrometry approach, based on peptidomics, offers the potential for integration into clinical labs already equipped with triple quadrupole mass spectrometry. Although mass spectrometry involves an initial capital investment, the per-sample reagent cost is only \$0.5 based on the listing price of all reagents and supplies, making the approach cost-effective for routine diagnostics.

The protocol is simple and suitable for high-throughput and automation, making it easy to integrate into daily clinical practice and scale up. Initial training is straightforward and does not require significant additional resources, further enhancing its scalability and ease of implementation.

Here is the new discussion we have added in the revised manuscript in **Page 13, Paragraph 2:** “PEP-TORCH eliminates the need for extended culture times (save averaging 13 days in our validation set), significantly accelerating diagnosis and treatment. It is being developed into a

user-friendly, one-click software, streamlining data processing and reducing the need for experienced technicians, which facilitates seamless integration into clinical laboratories. While TB remains a challenge in resource-limited settings, the rise in NTM infections in developed countries underscores the relevance of PEP-TORCH for centralized labs in these regions. Our targeted mass spectrometry approach, leveraging peptidomics, aligns well with clinical labs already equipped with triple quadrupole mass spectrometry. Although mass spectrometry requires an initial capital investment, the low per-sample reagent cost ensures cost-effectiveness for routine use. The simple protocol supports high-throughput applications, requiring minimal training and resources, further enhancing its scalability and suitability for clinical adoption.”

Q3: While the figures are helpful, adding a summary table comparing PEP-TORCH's performance metrics with other methods would enhance readability

Response Q3: Thank you for the suggestion. In the original manuscript, we compared PEP-TORCH's performance metrics with existing clinical methods in **Table 2 and Table 3** and have updated these tables to incorporate data from 29 additional samples. We have also revised **Figure 4A** to account for these additional samples.

Table 2. Comparison of PEP-TORCH results with reference identification.

ID reference	Species IDs			Subspecies IDs		
	MALDI (Sequencing)*	PEP- TORCH	Agreement	WGS	PEP- TORCH	Agreement
M. tuberculosis Complex						
M. tuberculosis	24/26 (2/26)*	26/26	100%			
M. bovis				2/2	2/2	100%
M. bovis BCG				1/1	1/1	100%
M. africanum				2/2	2/2	100%
NTM						
M. abscessus	30/30	30/30	100%			
M. abscessus subsp. abscessus				7/7	7/7	100%
M. abscessus subsp. massiliense				5/5	5/5	100%
M. kansasii	1/7 (6/7)*	7/7	100%			
M. avium	7/7	7/7	100%			
M. intracellulare/Chimaera	5/5	5/5	100%			
Co-infection	1/1	1/1	100%			
M. goodii	1/1	1/1	100%			
M. mucogenicum	1/1	1/1	100%			
M. fortuitum	1/1	1/1	100%			
M. simiae **	1/1	1/1	100%			
M. florentinum **	1/1	1/1	100%			
Total	81/81	81/81	100%	17/17	17/17	100%

*, Sequencing is performed either by polymerase chain reaction or whole genome sequencing

***, The isolates were received from The BEI Resources Repository. The catalog No. were NR-4434 (*M.simiae*) and NR-49073 (*M.florentinum*).

Table 3. Peptide targets validated in mycobacterial species by parallel reaction monitoring mass-spectrometry

ID reference	Samples			Validation by PRM	Agreement
	Sample sub-set from Table 2	New sample set	Total		
M. tuberculosis	14/14	0	14	14/14	100%
M. abscessus	16/16	3/3	19	19/19	100%
M. kansasii	6/6	1/1	7	7/7	100%
M. avium	3/3	11/11	14	14/14	100%
M. intracellulare	3/3	6/6	9	9/9	100%
Total Samples used in PRM	42	21	63	63/63	100%

We also evaluated the time savings achieved by PEP-TORCH, primarily due to the elimination of the solid culture procedure. In contrast, the solid culture time is required for MALDI-Biotyper detection, the current standard method in clinical laboratories. The time saved ranged from 1 to 39 days, with an average of 13 days across all samples. The new **Figure 4B** illustrates the average days saved for different species and overall.

Figure 4. (A) Overall summary of sample identification by PEP-TORCH and other clinical tools at species and subspecies levels. Samples were identified using matrix-assisted laser desorption ionization time-of-flight (MALDI-TOF) mass spectrometry, polymerase chain reaction (PCR, Accuprobe), or whole genome sequencing (WGS). PEP-TORCH provided

perfect matches to all these clinical methods. **(B)** Solid culture time saved by the PEP-TORCH method compared to MALDI-TOF for different species. Species abbreviations: *M. abscessus* (Mab), *M. avium* complex (MAC), *M. tuberculosis* (Mtb), and *M. kansasii* (Mkan). The boxplot shows the median (middle line), maximum (upper limit), and minimum (lower limit) values. **(C-H)** The PEP-TORCH methodology assigned the peptide weightage (PW) scores for species (PW_{sp}) to facilitate their identities. All samples **(C)** attributed to *M. abscessus* by PEP-TORCH PW_{sp} scores were subsequently corroborated as *M. abscessus* through MALDI-MS analysis. Furthermore, subspecies PW scores (PW_{subsp}) were assigned to these samples, distinguishing them as either *M. abscessus* subspecies *abscessus* or *M. abscessus* subspecies *massiliense*. Notably, PEP-TORCH exhibited the capability to discern subspecies identities for all samples, however, only 17 samples were verifiably confirmed through WGS analysis, owing to the intricate and resource-intensive nature of WGS procedures. The PEP-TORCH analysis yielded PW_{sp} scores for samples identified as belonging to various species: **(D)** *M. avium* with PW_{subsp} , and **(E)** *M. intracellulare* with PW_{subsp} . **(F)** *M. kansasii*, **(G)** One case was diagnosed as co-infected by *M. kansasii* / *M. avium* (TKK-01-0059, a strain of *M. avium*, from the PEP-TORCH results), and **(H)** Less common species including *M. simiae*, *M. florentinum*, *M. gordonae*, *M. fortuitum*, and *M. mucogenicum*. Heatmaps indicate differences in PW scores with dark red at 100, red between 10-100, and blue was less than 10.

Q4: Figure 4 "C M. Kansasii" should say "M. kansasii".

Response Q4: We apologize for the confusion. The "C" in "C M. Kansasii" refers to Panel C of Figure 4. To clarify, we have revised the formatting of Figure 4 to clarify this point.

Q5: There are a few typographical errors that should be corrected (e.g., "The PEP-TORCH algorithm enhances mycobacterial classification by accuracy identifying species..." should read "accurately identifying species").

Response Q5: We regret these errors and have carefully proof-read and revised the manuscript to correct them.

Conclusion: This manuscript provides a valuable contribution to the field of microbiological diagnostics and could significantly impact the rapid identification of mycobacterial infections.

Referee #3 (Comments on Novelty/Model System for Author):

Dr Fan and colleagues develop a novel LC-MS/Algorithm capable of identifying NTM/Mtb species and sub-species. This novel technology platform is very important for clinical applications of diagnosis of NTM/Mtb. The accuracy of proof-of-concept application of such system in medium-size participants recruited from Florida are pretty impressive. Manuscript is well, clearly written.

Referee #3 (Remarks for Author):

I only have couple of minor comments for authors:

Q1: They focus on several key NTM species to develop and validate the system. However, can they discuss how this system may be potentially used to identify more NTM species in clinical?

Response Q1: The database used for peptide identification was downloaded from UniProtKB and targeted taxonomy ID:1762 (Mycobacteriaceae). It contained 2,084,438 entries from 7,108 descendant taxonomic groups, providing broad coverage of various species and strains. Mass spectrometry data were searched against this protein database to identify peptides. The identified peptides were then analyzed using PEP-TORCH, which accounts for all NTM species reported to infect humans, as per the Bacterial and Viral Bioinformatics Resource Center (<https://www.bv-brc.org/>).

Thus, our pipeline provides comprehensive coverage of a broad range of human-infecting NTM species. However, during validation, the majority of samples were from high-prevalence NTM species. For the revision, we successfully tested a total of five less prevalent NTM species (beyond the top four).

The main results from these updates are reflected in the revised **Figure 4** and **Tables 1–3**.

New description added to the revised manuscript in **Discussion, Page 13, Last Paragraph:** “*The pipeline is designed to detect a broader range of human-infecting NTM species. While this study primarily validated high-prevalence species, we also included five less common species. Future research will focus on further validation with additional less common species.*”

Q2: Can they discuss more about the power of this system for differential diagnosis between NTM and BCG vaccine strain since most of people in developing countries received BCG vaccination?

Response Q2: Thank you for the suggestion. The BCG vaccine is derived from an attenuated *M. bovis* strain with CFP-10 and ESAT-6 virulence factors deleted. As a member of the *M. tuberculosis* complex, *M. bovis* can be effectively distinguished by PEP-TORCH, which identifies species-specific peptides to differentiate between NTM and *M. tuberculosis* complex, including vaccine-derived strains.

In our study, 7 out of 26 MTB-positive cases had a history of BCG vaccination. Typically, individuals vaccinated with BCG do not develop active TB, so *M. tuberculosis* antigen peptides are not expected in their clinical samples. However, *M. tuberculosis* peptides may still be detected in two scenarios:

- Subsequent infection with *M. tuberculosis* leading to active TB.
- Rare cases of disseminated BCG infection (BCGosis), often occurring in immunocompromised individuals.

We encountered one case of disseminated TB, which is detailed in the original manuscript. Using the PRM-targeted method, we did not detect targeted *M. tuberculosis* peptides, as they were derived from CFP-10 and ESAT-6, which are absent in *M. bovis* BCG. However, the DDA-based PEP-TORCH workflow successfully identified other *M. tuberculosis* antigen peptides in this case.

In summary, our assay effectively differentiates NTM and *M. tuberculosis*, including cases with prior BCG vaccination, through its robust species-specific peptide identification approach.

We have added more discussion in this point in the revised manuscript **Page 11, Paragraph 1:** “*This vaccine, derived from an attenuated M. bovis strain that lacks CFP-10 and ESAT-6, typically does not cause the detection of M. tuberculosis peptides in vaccinated individuals, unless they are later infected with M. tuberculosis or, in rare cases, develop BCGosis.*”

Q3: Can they include the information for participants in Table 1 to more clearly show whether they received BCG vaccine?

Response Q3: We have added this BCG vaccine information to the legend of **Table 1**.

Table 1. Demographic information of the clinical samples.

Sample source	Male		Female		Total	
	n	Age Mean (min-max)	n	Age Mean (min-max)	n	Age Mean (min-max)
Sputum	30	52 (17-87)	39	54 (16-78)	73	54 (16-87)
Abscess	1	44	2	39 (35-44)	3	41 (35-44)
Biopsy lung	1	78			1	78
Biopsy lymph node	1	27	2	50 (41-60)	3	43 (27-60)
Biopsy skin	5	44 (23-61)			5	44 (23-61)
Bronchial wash	2	47 (46-49)	3	65 (46-80)	5	58 (46-80)
CSF			1	61	1	61
Right eye anterior chamber	1	78			1	78
Sinus	1	53			1	53
Stool			1	10	1	10
Bone marrow	1	40			1	40
Wound	3	23 (23-23)			3	23 (23-23)
Other Tissues	2	27 (23-40)			2	27 (23-40)
BCG Vaccinated*	1	27	6	31 (10-46)	7	30 (10-46)
Total	48	49 (17-87)	48	52 (16-80)	96	51 (16-87)

*, The detailed information on ‘BCG vaccinated’ was in Dataset EV1A. Only *M. tuberculosis* cases have the BCG vaccination record.

Q4: Can they discuss more about how this system may potentially avoid the interference from other bacteria (particularly anaerobic bacteria) colonized in lungs or other extra-pulmonary organs such as intestine because anaerobic bacteria from lungs or intestine may not be easily grown by protocol shown in Figure 1. Or more generally speaking, can they discuss how this system may avoid the interference from lung/gut microbiota?

Response Q4: One step in PEP-TORCH involves matching all identified peptides through Unipept to generate a species-match matrix. Species not documented as mycobacteria isolated

from human hosts are removed from the matrix. PEP-TORCH then identifies mycobacteria species-specific peptide events, as our new Appendix Table S2 demonstrates. These events represent peptides or 2-3 peptide combinations that uniquely match NTM or MTB species, excluding other taxa. Only the peptides or peptide combinations that specifically represent a mycobacteria species are recognized by PEP-TORCH as valid events. For example, PEP-TORCH identified a unique peptide specific to *M. abscessus*. Below is a screenshot (**Response Figure 3**) of the lineage tree from UniPept, illustrating that this peptide uniquely matches *M. abscessus*.

Reponses Figure 3. UniPept platform output for taxonomy identification.

Reference

- Conyers LE, Saunders BM (2024) Treatment for non-tuberculous mycobacteria: challenges and prospects. *Frontiers in Microbiology* 15: 1394220
- Dartois V, Dick T (2024) Therapeutic developments for tuberculosis and nontuberculous mycobacterial lung disease. *Nature Reviews Drug Discovery* 23: 381-403
- Dumais MG, Wengenack NL, Norgan AP, Amin S, Sia IG, Rhee PC, Connelly BJ, Arment CA (2023) Toto, we're not in Kansas anymore: First reported case of *M. persicum* septic arthritis. *Journal of Clinical Tuberculosis and Other Mycobacterial Diseases* 31: 100352
- Menzies NA, Quaipe M, Allwood BW, Byrne AL, Coussens AK, Harries AD, Marx FM, Meghji J, Pedrazzoli D, Salomon JA (2021) Lifetime burden of disease due to incident tuberculosis: a global reappraisal including post-tuberculosis sequelae. *The Lancet Global Health* 9: e1679-e1687
- Nakamura K, Murakami E, Kishino D, Mashimo S, Kurioka Y, Shibata Y, Taniguchi A, Higo H, Hiramatsu Y, Maeda Y (2023) Mycobacterium shinjukuense infection successfully treated with clarithromycin, rifampicin, and ethambutol. *Respiratory Medicine Case Reports* 45: 101894
- Perandones C, Roncoroni A, Frega N, Bianchini H, Hübscher O (1991) Mycobacterium gastritis: septic arthritis due to Mycobacterium gastritis in a patient with a renal transplant. *The Journal of Rheumatology* 18: 777-778
- Riopel ND, Long R, Heffernan C, Tyrrell GJ, Shandro C, Li V, Islam MR, Stobart M, Sharma MK, Soualhine H (2024) Characterization of Mycobacterium orygis, Mycobacterium bovis, and Mycobacterium caprae Infections in Humans in Western Canada. *The Journal of Infectious Diseases*: jiae124
- Shahraki AH, Trovato A, Mirsaedi M, Borroni E, Heidarieh P, Hashemzadeh M, Shahbazi N, Cirillo DM, Tortoli E (2017) Mycobacterium persicum sp. nov., a novel species closely related to Mycobacterium kansasii and Mycobacterium gastritis. *International Journal of Systematic and Evolutionary Microbiology* 67: 1766-1770
- Simmon KE, Brown-Elliott BA, Ridge PG, Durtschi JD, Mann LB, Slechta ES, Steigerwalt AG, Moser BD, Whitney AM, Brown JM (2011) Mycobacterium chelonae-abscessus complex associated with sinopulmonary disease, Northeastern USA. *Emerging infectious diseases* 17: 1692

Somoskovi A, Dormandy J, Mayrer AR, Carter M, Hooper N, Salfinger M (2009) "Mycobacterium canettii" isolated from a human immunodeficiency virus-positive patient: first case recognized in the United States. *Journal of clinical microbiology* 47: 255-257

Tateishi Y, Ozeki Y, Nishiyama A, Miki M, Maekura R, Fukushima Y, Nakajima C, Suzuki Y, Matsumoto S (2021) Comparative genomic analysis of Mycobacterium intracellulare: implications for clinical taxonomic classification in pulmonary Mycobacterium avium-intracellulare complex disease. *BMC microbiology* 21: 1-15

Velayati AA, Boloorsaze MR, Farnia P, Mohammadi F, Karam MB, Masjedi MR (2005) Mycobacterium gastri causing disseminated infection in children of same family. *Pediatric pulmonology* 39: 284-287

Wang Y, Boulic M, Phipps R, Plagmann M, Cunningham C, Guyot G (2023) Field performance of a solar air heater used for space heating and ventilation—A case study in New Zealand primary schools. *Journal of Building Engineering* 76: 106802

Zurita J, Ortega-Paredes D, Mora M, Espinel N, Parra H, Febres L, Zurita-Salinas C (2014) Characterization of the first report of Mycobacterium timonense infecting an HIV patient in an Ecuadorian hospital. *Clinical Microbiology and Infection* 20: O1113-O1116

4th Feb 2025

Dear Dr. Fan,

Thank you for the submission of your revised manuscript to EMBO Molecular Medicine. We have now heard back from the one referee who agreed to re-evaluate your manuscript. This referee also assessed author responses to concerns raised by other referees. I have carefully read your manuscript, point-by-point response to the referees' comments, referee report and discussed it with the other members of our editorial team. I am pleased to inform you that we will be able to accept your manuscript pending the following final amendments:

- 1) Please implement all suggestions made by the referee.
- 2) We note that you currently have, a total of 3 first authors. Is that correct? Do you confirm equal contribution of these authors, able to take full responsibility for the paper and its content? While there is no limit per se to the number of first authors, 3 authors is rather rare, and may not reflect as intended to the community.
- 3) In the main manuscript file, please do the following:
 - Please address all comments suggested by our data editors listed below:
 - o Data availability statement:
 1. Please note that the specific URL for PXD059923 dataset is not provided in the data availability statement.
 - o Figure legends:
 1. Please note that the box plots need to be defined in terms of bounds of box and whiskers, and percentile in the legend of figure 4B.
 2. Please note that information related to n is missing in the legends of figures 2E, 4B.
 - Add callouts for Fig 3B and Fig 4E. Also, please correct callouts of supplementary tables in lines 542 and 543.
 - Remove all figures and only leave their legends at the end of the file. Place tables after the EV figure legends.
 - Author contributions: Please remove it from the manuscript and specify author contributions in our submission system. CRediT has replaced the traditional author contributions section because it offers a systematic machine-readable author contributions format that allows for more effective research assessment. You are encouraged to use the free text boxes beneath each contributing author's name to add specific details on the author's contribution. More information is available in our guide to authors:
<https://www.embopress.org/page/journal/17574684/authorguide#authorshipguidelines>
 - In Methods, add the following paragraph:

Graphics:

(some of the... OR Figure #... OR synopsis) Graphics were created with BioRender.com.

- In Methods, provide the statement that informed consent was obtained from all human subjects and confirm that the experiments conformed to the principles set out in the WMA Declaration of Helsinki and the Department of Health and Human Services Belmont Report. Please enter this information also in the Author Checklist.
- In Methods, add a statistical paragraph that should reflect all information that you have filled in the Authors Checklist, especially regarding randomization, blinding, replication.
- Indicate in legends number and nature of replicates and exact p= values, not a range, along with the statistical test used. To keep the figures "clear" some authors found providing an Appendix table Sx with all exact p-values preferable. You are welcome to do this if you want to.
- Data availability: Make sure that all data deposited in public repositories are freely accessible upon publication. Also, please check provided URLs and make sure that the link is correct.

Use the following format to report the accession number of your data:

[data type]: [full name of the resource] [accession number/identifier] ([doi or URL or identifiers.org/DATABASE:ACCESSION])

Please check "Author Guidelines" for more information.

<https://www.embopress.org/page/journal/17574684/authorguide#availabilityofpublishedmaterial>

4) Datasets: Please remove dataset legends from the manuscript text and add them to the corresponding dataset files, in a separate tab/worksheet. Please add a short description of the datasets.

5) Synopsis:

- Synopsis image: Please submit the visual abstract as a separate, high-resolution jpeg file 550 pixels wide x 200-600 pixels high.
- Please check your synopsis text and image before submission with your revised manuscript. Please be aware that in the proof stage minor corrections only are allowed (e.g., typos).

6) As part of the EMBO Publications transparent editorial process initiative (see our Editorial at

<http://embomolmed.embopress.org/content/2/9/329>), EMBO Molecular Medicine will publish online a Review Process File (RPF) to accompany accepted manuscripts. This file will be published in conjunction with your paper and will include the anonymous

referee reports, your point-by-point response and all pertinent correspondence relating to the manuscript. Let us know whether you agree with the publication of the RPF and as here, if you want to remove or not any figures from it prior to publication. Please note that the Authors checklist will be published at the end of the RPF.

7) Please provide a point-by-point letter INCLUDING my comments as well as the reviewer's reports and your detailed responses (as Word file).

I look forward to reading a new revised version of your manuscript as soon as possible.

Yours sincerely,

Zeljko Durdevic

*** Instructions to submit your revised manuscript ***

1) a .docx formatted version of the manuscript text (including Figure legends and tables)

2) Separate figure files*

3) supplemental information as Expanded View and/or Appendix. Please carefully check the authors guidelines for formatting Expanded view and Appendix figures and tables at <https://www.embopress.org/page/journal/17574684/authorguide#expandedview>

4) a letter INCLUDING the reviewer's reports and your detailed responses to their comments (as Word file).

5) The paper explained: EMBO Molecular Medicine articles are accompanied by a summary of the articles to emphasize the major findings in the paper and their medical implications for the non-specialist reader. Please provide a draft summary of your article highlighting

6) Author contributions: the contribution of every author must be detailed in a separate section.

7) EMBO Molecular Medicine now requires a complete author checklist (<https://www.embopress.org/page/journal/17574684/authorguide>) to be submitted with all revised manuscripts. Please use the checklist as guideline for the sort of information we need WITHIN the manuscript. The checklist should only be filled with page numbers where the information can be found. This is particularly important for animal reporting, antibody dilutions (missing) and

exact values and n that should be indicated instead of a range.

8) Every published paper now includes a 'Synopsis' to further enhance discoverability. Synopses are displayed on the journal webpage and are freely accessible to all readers. They include a short stand first (maximum of 300 characters, including space) as well as 2-5 one sentence bullet points that summarise the paper. Please write the bullet points to summarise the key NEW findings. They should be designed to be complementary to the abstract - i.e. not repeat the same text. We encourage inclusion of key acronyms and quantitative information (maximum of 30 words / bullet point). Please use the passive voice. Please attach these in a separate file or send them by email, we will incorporate them accordingly.

You are also welcome to suggest a striking image or visual abstract to illustrate your article. If you do please provide a jpeg file 550 px-wide x 300-600px high.

9) A Conflict of Interest statement should be provided in the main text

10) Please note that we now mandate that all corresponding authors list an ORCID digital identifier. This takes <90 seconds to complete. We encourage all authors to supply an ORCID identifier, which will be linked to their name for unambiguous name identification.

Currently, our records indicate that the ORCID for your account is 0000-0003-3384-0834.

Link Not Available

11) Include a Reagents and Tools Table as part of the Methods section, which can be downloaded from our author guidelines (<https://www.embopress.org/page/journal/17574684/authorguide#structuredmethods>)

Photos 400-800 DPI

*Additional important information regarding figures and illustrations can be found at

<https://bit.ly/EMBOPressFigurePreparationGuideline>. See also figure legend preparation guidelines:

<https://www.embopress.org/page/journal/17574684/authorguide#figureformat>

***** Reviewer's comments *****

Referee #3 (Comments on Novelty/Model System for Author):

They already used highly-relevant clinical samples to test their method. Thus, model system is adequate.

Referee #3 (Remarks for Author):

Authors have addressed most of my previous comments , I only have several minor comments that can be quickly addressed by Authors, and this work should be acceptable:

1. In line 51, "confirming their suitability as biomarkers in clinical settings". "confirming" is somewhat too strong as no large-scale, multi-center trial in compliance with golden standard is done yet. Should be please changed to "suggesting " or "indicating"?

2. "tuberculosis" in line 847 should be "TB". Also, other Mycobacterial species names should be italic in this figure legend. Please double check.

3. In line 442, a discussion regarding the financial feasibility of the clinical application potential of PEP-TORCH was included during the revision. It is somewhat redundant to mention this for a scientific paper. Actually, the cost of LC-MS/MS tests is (should be) getting cheaper in the near future. However, the real power of PEP-TORCH may be the potential of combined applications of "PEP-TORCH" with application of LC-MS/MS for detecting the metabolites or other biomarkers during mycobacterial infection. Thus, can author please add one or two more sentences in limitation discussion paragraph in line 458 like this: "Also, the integrative applications of PEP-TORCH with other mass spectrometry-driving tests of mycobacteria-specific or -associated metabolites (Cite: Chen et al, Nature Metabolism,4 (3), 359-373, 2022.) or other cutting-edge biomarkers (Cite: Das et al, ACS Nano, 18 (14), 2024; Maclean et al, Nature Microbiology, 4, 748-758,2019) should open a new avenue for

diagnosis with higher specificity and accuracy for mycobacterial infections with additional less common species or complex infection settings such as HIV/AIDS or Mtb/NTM co-infections."

***** Reviewer's comments *****

Referee #3 (Comments on Novelty/Model System for Author):

They already used highly-relevant clinical samples to test their method. Thus, model system is adequate.

Referee #3 (Remarks for Author):

Authors have addressed most of my previous comments , I only have several minor comments that can be quickly addressed by Authors, and this work should be acceptable:

1.In line 51, "confirming their suitability as biomarkers in clinical settings". "confirming" is somewhat too strong as no large-scale, multi-center trial in compliance with golden standard is done yet. Should be please changed to "suggesting " or "indicating"?

Response: Thank you for your thorough review. We have replaced “confirming” with “indicating” in **Line 51** of the revised manuscript.

2."tuberculosis" in line 847 should be "TB". Also, other Mycobacterial species names should be italic in this figure legend. Please double check.

Response: Thank you for your careful review. We have updated the figure legend and the main text to ensure all Mycobacterial species and subspecies names are italicized, and we have changed "tuberculosis" to "TB" in **Line 822** of the revised manuscript.

3.In line 442, a discussion regarding the financial feasibility of the clinical application potential of PEP-TORCH was included during the revision. It is somewhat redundant to mention this for a scientific paper. Actually, the cost of LC-MS/MS tests is (should be) getting cheaper in the near future. However, the real power of PEP-TORCH may be the potential of combined applications of "PEP-TORCH" with application of LC-MS/MS for detecting the metabolites or other biomarkers during mycobacterial infection. Thus, can author please add one or two more sentences in limitation discussion paragraph in line 458 like this: "Also, the integrative applications of PEP-TORCH with other mass spectrometry-driving tests of mycobacteria-specific or -associated metabolites (Cite: Chen et al, Nature Metabolism,4 (3), 359-373, 2022.) or other cutting-edge biomarkers (Cite: Das et al, ACS Nano, 18 (14), 2024; Maclean et al, Nature Microbiology, 4, 748-758,2019) should open a

new avenue for diagnosis with higher specificity and accuracy for mycobacterial infections with additional less common species or complex infection settings such as HIV/AIDS or Mtb/NTM co-infections."

Response: Thank you for your suggestions on the discussion section. We have incorporated the suggested descriptions and citations into **Line 459-564** of the revised manuscript.

19th Feb 2025

Dear Dr. Fan,

We are pleased to inform you that your manuscript is accepted for publication and is now being sent to our publisher to be included in the next available issue of EMBO Molecular Medicine.

Zeljko Durdevic
Senior Editor
EMBO Molecular Medicine
